# Structural model of microtubule dynamics inhibition by kinesin-4 from the crystal structure of KLP-12 –tubulin complex

**Shinya Taguchi**[1,2†], **Juri Nakano**[3†], **Tsuyoshi Imasaki**[1†], **Tomoki Kita**[4], **Yumiko Saijo-Hamano**[1], **Naoki Sakai**[5], **Hideki Shigematsu**[5], **Hiromichi Okuma**[1], **Takahiro Shimizu**[1], **Eriko Nitta**[1], **Satoshi Kikkawa**[1], **Satoshi Mizobuchi**[2], **Shinsuke Niwa**[3,4,6]*, **Ryo Nitta**[1]*

[1]Division of Structural Medicine and Anatomy, Department of Physiology and Cell Biology, Kobe University Graduate School of Medicine, Kobe, Japan; [2]Division of Anesthesiology, Kobe University Graduate School of Medicine, Kobe, Japan; [3]Graduate School of Life Sciences, Tohoku University, Sendai, Japan; [4]Department of Applied Physics, Graduate School of Engineering, Tohoku University, Sendai, Japan; [5]RIKEN SPring-8 Center, Sayo, Japan; [6]Frontier Research Institute for Interdisciplinary Sciences (FRIS), Tohoku University, Sendai, Japan

**\*For correspondence:**
shinsuke.niwa.c8@tohoku.ac.jp (SN);
ryonitta@med.kobe-u.ac.jp (RN)

†These authors contributed equally to this work

**Abstract** Kinesin superfamily proteins are microtubule-based molecular motors driven by the energy of ATP hydrolysis. Among them, the kinesin-4 family is a unique motor that inhibits microtubule dynamics. Although mutations of kinesin-4 cause several diseases, its molecular mechanism is unclear because of the difficulty of visualizing the high-resolution structure of kinesin-4 working at the microtubule plus-end. Here, we report that KLP-12, a *C. elegans* kinesin-4 ortholog of KIF21A and KIF21B, is essential for proper length control of *C. elegans* axons, and its motor domain represses microtubule polymerization in vitro. The crystal structure of the KLP-12 motor domain complexed with tubulin, which represents the high-resolution structural snapshot of the inhibition state of microtubule-end dynamics, revealed the bending effect of KLP-12 for tubulin. Comparison with the KIF5B-tubulin and KIF2C-tubulin complexes, which represent the elongation and shrinking forms of microtubule ends, respectively, showed the curvature of tubulin introduced by KLP-12 is in between them. Taken together, KLP-12 controls the proper length of axons by modulating the curvature of the microtubule ends to inhibit the microtubule dynamics.

## Editor's evaluation

In their study, Taguchi et al. aim to determine how a member of the kinesin-4 family is able to stabilize the tips of microtubules to suppress both their growth and shrinkage, a process important for normal development. This paper provides convincing data on KLP-12 by combining in vivo *C. elegans* work with in vitro single-molecule analysis and structural studies of the motor domain. The structure shows that KLP-12 bends tubulin heterodimers to a level that lies in between the extremes of bending by KIF5B (lattice stabilizer) and KIF2C (lattice destabilizer). This important study will be of interest to those in the fields of neuronal development and cytoskeletal dynamics.

## Introduction

Kinesin superfamily proteins (KIFs) are microtubule-based molecular motors driven by the energy of ATP hydrolysis (*Hirokawa et al., 2009a*). Most KIFs move along microtubules to transport various

**eLife digest** From meter-long structures that allow nerve cells to stretch across a body to miniscule 'hairs' required for lung cells to clear mucus, many life processes rely on cells sporting projections which have the right size for their role. Networks of hollow filaments known as microtubules shape these structures and ensure that they have the appropriate dimensions. Controlling the length of microtubules is therefore essential for organisms, yet how this process takes place is still not fully elucidated.

Previous research has shown that microtubules continue to grow when their end is straight but stop when it is curved. A family of molecular motors known as kinesin-4 participate in this process, but the exact mechanisms at play remain unclear.

To investigate, Tuguchi, Nakano, Imasaki et al. focused on the KLP-12 protein, a kinesin-4 equivalent which helps to controls the length of microtubules in the tiny worm *Caenorhabditis elegans*. They performed genetic manipulations and imaged the interactions between KLP-12 and the growing end of a microtubule using X-ray crystallography. This revealed that KLP-12 controls the length of neurons by inhibiting microtubule growth. It does so by modulating the curvature of the growing end of the filament to suppress its extension. A 'snapshot' of KLP-12 binding to a microtubule at the resolution of the atom revealed exactly how the protein helps to bend the end of the filament to prevent it from growing further.

These results will help to understand how nerve cells are shaped. This may also provide insights into the molecular mechanisms for various neurodegenerative disorders caused by problems with the human equivalents of KLP-12, potentially leading to new therapies.

cargos, including membranous organelles, protein complexes, and mRNAs (*Guedes-Dias and Holzbaur, 2019*; *Hirokawa et al., 2009b*). In addition to transporting cargos, some kinesins possess the ability to regulate microtubule dynamics, such as elongation (polymerization), catastrophe or shrinkage (depolymerization), in diverse ways. Kinesin-1, the founding member of the KIFs, changes the conformation of unstable GDP microtubules into a conformation resembling the GTP microtubules (*Muto et al., 2005*; *Shima et al., 2018*). Conversely, kinesin-13 destabilizes both the plus and minus ends of microtubules to induce catastrophe or depolymerization (*Desai et al., 1999*; *Ogawa et al., 2004*). Kinesin-8 moves processively toward the microtubule plus-end, where it depolymerizes microtubule (*Gupta et al., 2006*; *Niwa et al., 2012*; *Varga et al., 2006*; *Wang et al., 2016*).

Kinesin-4, another family of kinesins, is known to inhibit microtubule dynamics and is classified into three subfamilies: the KIF4 subfamily, the KIF7 subfamily, and the KIF21 subfamily (*Yue et al., 2018*). The KIF21 subfamily has attracted considerable attention because its genetic alterations are linked with several diseases. Point mutations of KIF21A cause congenital fibrosis of the extraocular muscle type 1 (CFEOM1) (*Yamada et al., 2003*). Polymorphisms in the KIF21B gene are associated with several inflammatory diseases, such as multiple sclerosis or Crohn's disease (*Barrett et al., 2008*; *Goris et al., 2010*). Increased expression of KIF21B is also linked to the progression of neurodegenerative disorder (*Kreft et al., 2014*). *Kif21b* knockout mice were reported to exhibit behavioral changes involving impaired learning and memory (*Muhia et al., 2016*).

The molecular mechanisms of how kinesin-4 affects microtubule dynamics have been studied for more than a dozen years, demonstrating the main contribution of their motor domains to microtubule dynamics inhibition. Xklp1/KIF4, a fast processive motor, was first reported to reduce both microtubule growth and the catastrophe rate (*Bieling et al., 2010*; *Bringmann et al., 2004*). Its motor domain is able to bind not only to the microtubule lattices for microtubule-based motility but also to the curved tubulin dimers for inhibition of microtubule dynamics. Nonmotile KIF7 was reported to reduce the microtubule growth rate but enhance catastrophe to organize the tips of ciliary microtubules (*He et al., 2014*; *Yue et al., 2018*). The processive motor KIF21A/B reduces microtubule growth and catastrophes similar to Xklp1/KIF4 (*van der Vaart et al., 2013*; *van Riel et al., 2017*). These studies suggest that the motor domains of kinesin-4 family proteins play a crucial role in reducing the growth rate of microtubules. In other words, minor alterations in kinesin-4 motor domains affect their conserved functions to suppress microtubule dynamics by displaying strikingly distinct motility characteristics (*van der Vaart et al., 2013*; *van Riel et al., 2017*).

The other domains of kinesin-4 are also known to be involved in microtubule dynamics inhibition by regulating or supporting motor domain functions. The coiled-coil region of KIF21A in which CFEOM1-associated mutations are localized operates as an autoinhibitory domain by interaction with the motor domain (*Bianchi et al., 2016*; *Cheng et al., 2014*; *van der Vaart et al., 2013*). The dominant character of CFEOM1 syndrome is thus connected to the increased activity of the mutant KIF21A kinesin caused by the loss of autoinhibition. The WD40 domain of KIF21B holds on to the growing microtubule tip and induces its pausing (*van Riel et al., 2017*), which is required for the sustained action of the motor domain on the microtubule plus-end to inhibit microtubule dynamics.

We previously reported the first crystal structure of the KIF4 motor domain (*Chang et al., 2013*) and showed molecular mechanisms of ATP-induced motion; however, because of the lack of functional and structural information on kinesin-4 on the microtubule plus-end, the mechanism by which kinesin-4 motors inhibit microtubule dynamics is still obscure. Here, we investigated the functional and structural analyses of microtubule dynamics inhibition by kinesin-4 KLP-12, a *Caenorhabditis elegans (C. elegans)* ortholog of KIF21A and KIF21B (*Figure 1A*; *Figure 1—figure supplement 1*). Genetic analyses and in vitro TIRF (Total Internal Reflection Fluorescence microscopy) assays showed that KLP-12 regulates axonal length through inhibiting microtubule dynamics at its plus-end, similar to KIF21A and KIF21B. The crystal structure of KLP-12 complexed with curved α-, β- tubulin dimers suggested the structural model of microtubule dynamics inhibition by the kinesin-4 motor domain; kinesin-4 precisely controls the curvature of tubulin dimers at the plus-end, which is larger than that decorated by plus-end stabilizing kinesin-1 and smaller than that decorated by destabilizing kinesin-13. This precise control was achieved by the specific interactions on the microtubule interfaces conserved among kinesin-4 motors.

## Results

### KLP-12 regulates the length of axons in *C. elegans* neurons

KLP-12 is predicted a worm orthologue of KIF21A and KIF21B, which regulates axonal length. However, the function of KLP-12 remains to be elusive. Thus, we firstly analyzed the phenotype of *klp-12* mutants. We used two independent deletion mutant alleles of *klp-12*, *klp-12(tm10890)* and *klp-12(tm5176)* (*Figure 1B*). *klp-12(tm10890)* was considered to be a null allele because the mutation induces deletion of exon 4–6, resulting in a frameshift. *klp-12(tm5176)* had a deletion mutation in exons encoding the tail domain. We observed the development of two mechanosensory neurons, anterior lateral mechanosensory (ALM) and posterior lateral mechanosensory (PLM) neurons (*Figure 1C*) because the tiling between PLM and ALM neurons are strictly regulated by microtubule regulating factors, such as kinesin-13. Previous studies have shown worm mutants with more stable microtubules have defects in tiling between PLM and ALM (*Puri et al., 2021*). In wild-type animals, the axonal tip of PLM neurons does not reach the cell body of ALM without overlapping with each other (wild-type in *Figure 1D and E*; *Gallegos and Bargmann, 2004*). On the other hand, more than 30% of the PLM axons in *C. elegans* with *klp-12(tm5176)* overtook the cell body of ALM (*klp-12(tm5176)* in *Figure 1D and E*). *klp-12(tm10890)* showed a more severe phenotype; more than 50% of neurons overlapped, and thin warped axons were observed (*klp-12(tm10890)* in *Figure 1D and E*). We also investigated the effect of overexpression of wild-type KLP-12 in *C. elegans* neurons. Compared to the wild type, the PLM axon overexpressing KLP-12 became strikingly shorter and thinner (*Figure 1F and G*). Together with these results, the appropriate activity of KLP-12 is necessary to achieve proper length control of axons, suggesting that the function of KIF21/KLP-12 family proteins (*Figure 1A*) is evolutionarily conserved.

### KLP-12 is a plus-end directed motor that represses microtubule polymerization

Mammalian orthologs of KLP-12, KIF21A, and KIF21B, regulate the axon length by inhibiting the microtubule polymerization (*van der Vaart et al., 2013*; *van Riel et al., 2017*). Thus, KLP-12 may also inhibit the microtubule polymerization to restrict the length of axons. To directly visualize the KLP-12 function on the microtubules, we performed the in vitro TIRF assays. Since the neck-coiled-coil sequence of KLP-12 is not sufficient to form a stable dimer to display the processive motility in vitro, we intended to introduce the leucine zipper (LZ) of GCN4 and GFP right after KLP-12 (1-393) for

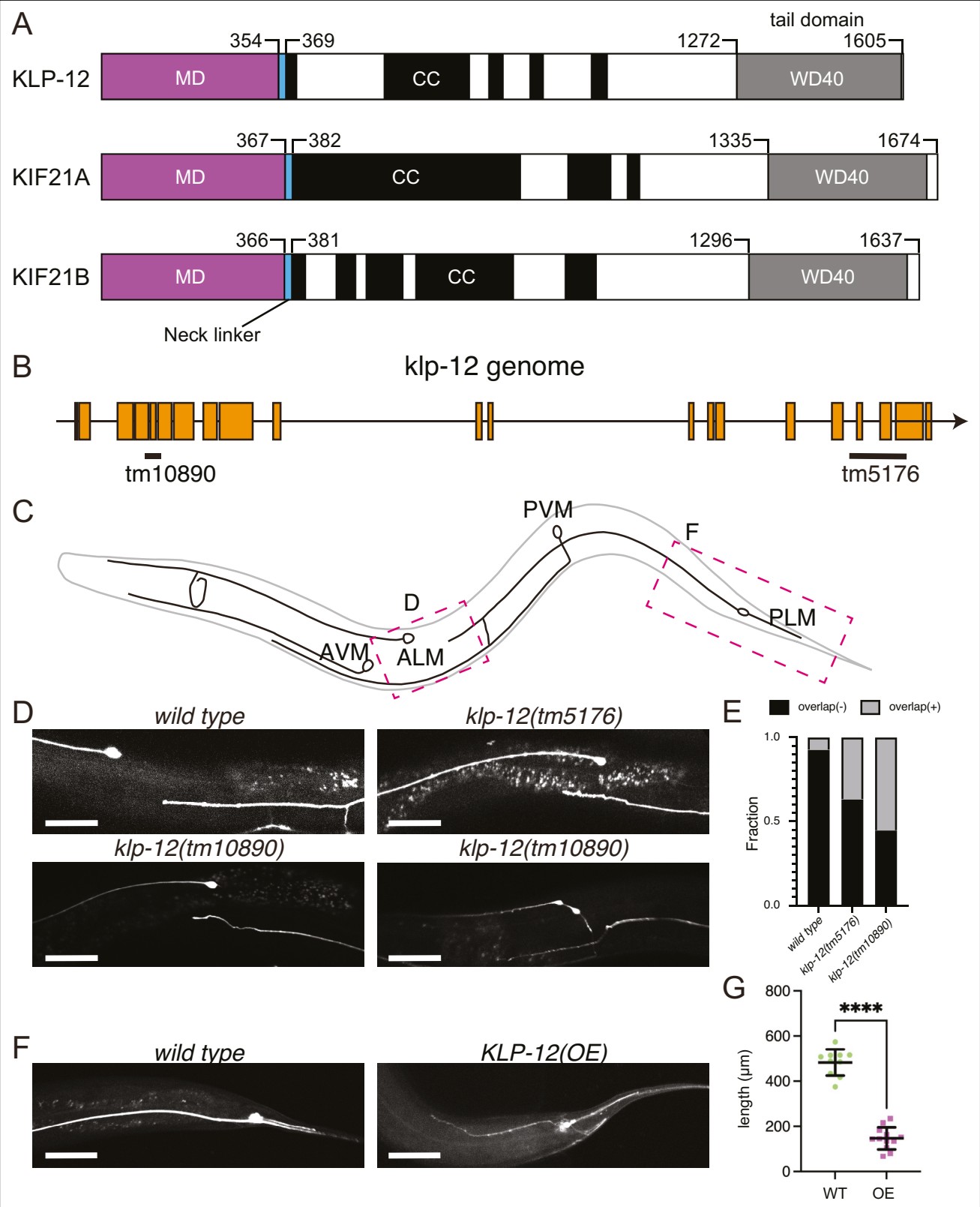

**Figure 1.** KLP-12 is an ortholog of KIF21A and KIF21B that regulates axonal length. (**A**) Schematic presentation of the domain organization of the KIF21 subfamily: *C. elegans* KLP-12, human KIF21A, and KIF21B consist of a motor domain (MD; magenta), neck linker (blue), coiled-coil domains (CC; black), and WD40 domain (WD40; gray). Phylogenetic tree and sequence alignment of kinesin-4 family are available in *Figure 1—figure supplement 1* and *Figure 1—figure supplement 2*, respectively. (**B**) Schematic presentation of the genomic structure of *C. elegans klp-12* and mutant alleles used

*Figure 1 continued on next page*

*Figure 1 continued*

in this study. tm10890 is a deletion mutant that induces frameshift, whereas tm5176 deletes WD repeats domain. (**C**) Schematic presentation of the mechanosensory neurons in *C. elegans*. Areas observed in panels (**D**) and (**F**) are shown by magenta boxes. ALM: anterior lateral mechanosensory, AVM: anterior ventral mechanosensory, PLM: posterior lateral mechanosensory, and PVM: posterior ventral mechanosensory neurons. (**D and E**) The tiling of ALM and PLM neurons. (**D**) Representative images of ALM and PLM neurons in *wild type* and *klp-12* mutant worms. The axonal tip of PLM neurons does not overlap with the cell body of ALM neurons in wild type, while the axonal tip of PLM neuron overlaps with the ALM cell body in *klp-12(tm5176)* and *klp-12(tm10890)* mutant alleles. Bars, 50 μm. (**E**) The percentage of ALM and PLM overlap in wild-type, *klp-12(tm5176)*, and *klp-12(tm10890)* mutant alleles in day 3 adult worms. n=55 in wild type, 60 in *klp-12(tm5176)*, and 58 in *klp-12(tm10890)* worms. (**F and G**) Overexpression of KLP-12 suppresses the elongation of mechanosensory axon. (**F**) The morphology of wild type and *klp-12*-overexpressed PLM neurites. Bar, 50 μm. (**G**) The lengths of PLM neurites are plotted. Each dots show the length of axons in each worm. Bars represent mean ± standard deviation. n=10 in wild-type and 12 in *KLP-12-expressing* axons. ****, p<0.0001, Welch's t test.

The online version of this article includes the following source data and figure supplement(s) for figure 1:

**Source data 1.** The length of neurons in wild-type (WT) and KLP-12-overexpressing (OE) *C. elegans*.

**Figure supplement 1.** Phylogenetic tree of representative kinesin-4 family member motor domains.

**Figure supplement 2.** The sequence alignment of kinesin-4 family proteins with KIF5B and KIF2C.

dimerization (*Tomishige et al., 2002*), which is a similar technique used in the previous kinesin-4 family motor study (*Yue et al., 2018*; *Figure 2A*: KLP-12–LZ–GFP). Single KLP-12–LZ–GFP motors successfully showed plus-end directed processive movements on GDP microtubules stabilized by paclitaxel (taxol-stabilized microtubules) with an average velocity of 0.81±0.32 μm/s and an average run length of 1.30±0.89 μm, which is a similar speed as the processive fast motor kinesin-1 or KIF4 (*Figure 2B–D*; *Shima et al., 2018*; *Yue et al., 2018*). The average velocity and run length of KLP-12–LZ–GFP motors were also similar on dynamic GTP microtubules (*Figure 2C–D*).

Next, we observed the microtubule dynamics and the localization of KLP-12–LZ–GFP at a series of different concentrations (*Figure 2E*). KLP-12–LZ–GFP did not accumulate the plus-end tips of microtubules but detached from them upon arrival of the tips, consistent with the previous results of KIF21A and KIF21B (*van der Vaart et al., 2013*; *van Riel et al., 2017*). KLP-12–LZ–GFP represented a concentration dependent decrease of the microtubule growth rate at its plus-end. In the absence of the KLP-12–LZ–GFP, the microtubule grew at the rate of 1.19±0.35 μm/min (*Figure 2F*). Increasing amounts of KLP-12–LZ–GFP inhibited the microtubule growth rate, with mean rates of 1.00 μm/min at concentrations of 30 nM and 0.77 μm/min at 300 nM, respectively (*Figure 2F*). On the other hand, microtubule depolymerization rate was not affected by KLP-12–LZ–GFP (*Figure 2H*). The frequencies of microtubule catastrophe events and rescue events were slightly reduced in the presence of KLP-12–LZ–GFP (*Figure 2G1*). It rises two possibilities; one is that KLP-12 reduces microtubule growth with a longer period of stabilization, and another is the indirect effect induced by reduced MT catastrophe events.

We further compared the inhibitory effect of KLP-12 with the previously reported dynamics with other kinesin-4 family motors, illustrating that the growth inhibition rate of KLP-12–LZ–GFP was very similar to that of KIF21A (*Figure 2—figure supplement 1*), a closely related family with 59.9% sequence identity to KLP-12 (*Figure 1—figure supplement 1*). These results showed that KLP-12–LZ–GFP possesses a suppression effect on the microtubule growth rates similar to the other members of the kinesin-4 family proteins, especially to KIF21A/B.

## The motor domain of KLP-12 binds to both the microtubule lattice and ends to catalyze ATP

Kinesin-4 moves along the microtubule until it reaches the plus-end, at which it inhibits the attachment and release of tubulin-dimers to/from the microtubule end. To achieve these dual functions, kinesin-4 must bind to the microtubule lattice and the microtubule end to catalyze ATP. We thus next focused on the monomeric motor domain of the KLP-12 (KLP-12(M)) (*Figure 2A*), investigating its ATPase activity in the presence of microtubules or tubulin heterodimers. We used GTP- and GDP- tubulin heterodimers to investigate the biochemical properties of KLP-12 at the plus-end of microtubules since the plus-end of microtubules is curved due to the lack of lateral interactions between protofilaments, as reported previously (*Hunter et al., 2003*).

Steady-state ATPase kinetics of KLP-12(M) were examined in the presence of taxol-stabilized microtubules (=microtubule lattice), GTP-tubulins (mimic of the growing microtubule end), and

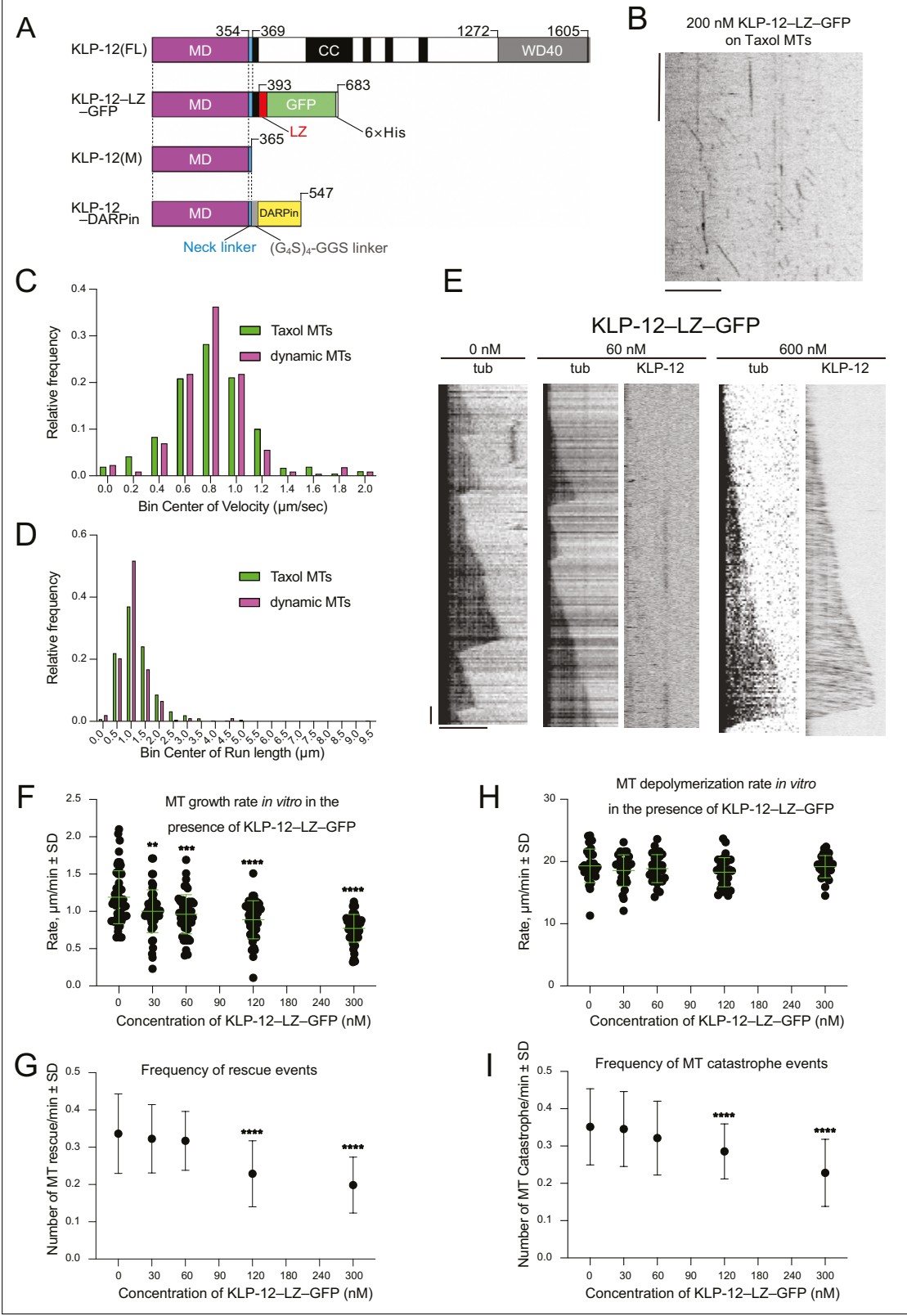

**Figure 2.** KLP-12 is a plus-end directed motor that represses microtubule polymerization. (**A**) Schematic presentation of the KLP-12 constructs. KLP-12(FL): full-length KLP-12, KLP-12–LZ–GFP: KLP-12 (1-393) with GFP connected with a leucine zipper, KLP-12(M): KLP-12 motor domain (1-365), KLP-12–DARPin: KLP-12(M) with DARPin connected with a flexible linker. (**B**) A representative kymograph showing the motility of KLP-12–LZ–GFP on taxol-stabilized microtubules. Horizontal and vertical bars show 10 μm and 10 s, respectively. (**C**) Histogram showing the velocity of KLP-12–LZ–GFP

*Figure 2 continued on next page*

*Figure 2 continued*

on taxol-stabilized (green) or dynamic (magenta) microtubules. 0.81±0.32 µm/s (n=407) and 0.82±0.31 µm/s (n=215) on taxol-stabilized and dynamic microtubules, respectively. Mean ± standard deviation. No statistically significant differences were detected by Student's t-test. (**D**) Histogram showing the run length of KLP-12–LZ–GFP on taxol-stabilized (green) or dynamic (magenta) microtubules. n=407 molecules. 1.30±0.89 µm (n=407) and 1.11±0.57 µm (n=215) on taxol-stabilized and dynamic microtubules, respectively. Mean ± standard deviation. No statistically significant differences were detected by Student's t-test. (**E**) Representative kymographs showing the microtubule dynamics and the motility of KLP-12–LZ–GFP. 10 µM of fluorescently labeled microtubules were polymerized from GMPCPP stabilized microtubule seeds fixed on the cover glass in the presence of 0, 60, or 600 nM KLP-12–LZ–GFP at 30 °C. Horizontal and vertical bars show 5 µm and 60 s, respectively. (**F–I**) The effect of KLP-12–LZ–GFP on microtubule dynamics. 10 µM of fluorescently labeled microtubules were observed in the presence of indicated concentrations of KLP-12–LZ–GFP at 30 °C. (**F**) Microtubule growth rate in vitro in the presence of KLP-12–LZ–GFP. Green bars show mean ± standard deviation. **, Adjusted p=0.0022, ***, Adjusted p=0.0001, ****, Adjusted p<0.0001, compared with control (0 nM). One-way ANOVA followed by Dunnett's multiple comparisons test. n=52 microtubules. (**G**) Frequency of microtubule catastrophe events. The number of microtubule catastrophe in vitro was normalized by minute. mean ± standard deviation. ****, Adjusted p<0.0001, compared with control (0 nM). One-way ANOVA followed by Dunnett's multiple comparisons test. n=101 microtubules. (**H**) Microtubule depolymerization rate in vitro in the presence of KLP-12–LZ–GFP. Green bars show mean ± standard deviation. No statistically significant differences were detected by One-Way ANOVA. n=30 microtubules. (**I**) Frequency of microtubule rescue events. The number of microtubule rescue events in vitro was normalized by minute. ****, Adjusted p<0.0001, compared with control (0 nM). One-way ANOVA followed by Dunnett's multiple comparisons test. n=99 microtubules. The effect of microtubule growth rate by kinesin-4 family motors is available in *Figure 2—figure supplement 1*.

The online version of this article includes the following source data and figure supplement(s) for figure 2:

**Source data 1.** Source data of microtubule growth rate in vitro in the presence of KLP-12–LZ–GFP (*Figure 2C, D, F, G, H and I*).

**Figure supplement 1.** The effect of microtubule growth rate by kinesin-4 family motors.

**Figure supplement 1—source data 1.** The ratio of the microtubule growth by kinesin-4 family motors.

GDP-tubulins (mimic of the shrinking microtubule end) (*Figure 3A*). The basal ATPase activity of KLP-12(M) in the absence of tubulins or microtubules was 0.0039±0.00095 s$^{-1}$, comparable with the other kinesin motors (*Hunter et al., 2003*; *Wang et al., 2016*). The ATPase activity of KLP-12(M) was stimulated ~33 times by microtubules to reach a maximum rate of 0.13 s$^{-1}$. The stimulation by free GTP- and GDP- tubulin dimers reached more than ~69 and~100 times to achieve a maximum rate of 0.27 and 0.39 s$^{-1}$, respectively. $K_{M,microtubules}$, $K_{M,GTP-tubulin}$, and $K_{M,GDP-tubulin}$ were 2.6 µM, 5.6 µM, and 22.2 µM, respectively (*Figure 3A*). These results indicate that KLP-12(M) binds similarly to the microtubules and the GTP-tubulin-dimers but shows considerably weaker binding to the GDP-tubulin heterodimers than to the microtubules or the GTP-tubulin heterodimers. We should note that the microtubule-activated ATPase rate was significantly lower than expected from the KLP-12–LZ–GFP velocity (*Figure 2B*). We thus examined the microtubule- and tubulin- activated ATPase rates of KLP-12–LZ–GFP, resulting in similar rates with those of KLP12(M), albeit ~ten times higher affinity to the tubulin or microtubule (*Figure 3—figure supplement 1*). We concluded that KLP-12 binds to both the microtubule lattice and the growing microtubule plus-end and activates its ATPase, thus achieving microtubule motility and growing microtubule stabilization. The discrepancy between the ATPase rate and the velocity is discussed in the Discussion section.

## Structure determination of the tubulin–KLP-12–DARPin complex

The binding ability of the KLP-12(M) to the soluble GTP-tubulin heterodimer was further analyzed by size exclusion chromatography (SEC). To prevent the self-assembly of tubulins, a designed ankyrin repeat protein (DARPin) that binds to the longitudinal interface of β-tubulin was used (*Ahmad et al., 2016*). Equimolar tubulin dimers, KLP-12(M) with the ATP analog AMP-PNP (adenylyl-imidodiphosphate), and DARPin were injected and analyzed by SEC representing three peaks (*Figure 3B*). The first peak consisted of all components, tubulin, KLP-12(M), and DARPin (*Figure 3C*). The second and third peaks correspond to KLP-12(M) and DARPin, respectively. KLP-12(M) and DARPin shifted to the left side to form a triple complex with tubulin. Thus, KLP-12(M) has a binding ability to a soluble GTP-tubulin heterodimer, consistent with steady-state ATPase assays.

Next, we proceeded to crystallize the KLP-12–GTP-tubulin complex to elucidate the molecular mechanism of microtubule stabilization by kinesin-4. For crystallization, to stabilize complex formation between KLP-12(M) and tubulin, KLP-12(M) was fused to DARPin by a long linker. According to a previous study (*Wang et al., 2017*), the linker length between the C-terminus of KLP-12(M) and the N-terminus of DARPin was optimized. Four G$_4$S repeats with one G$_2$S (KLP-12–DARPin fusion

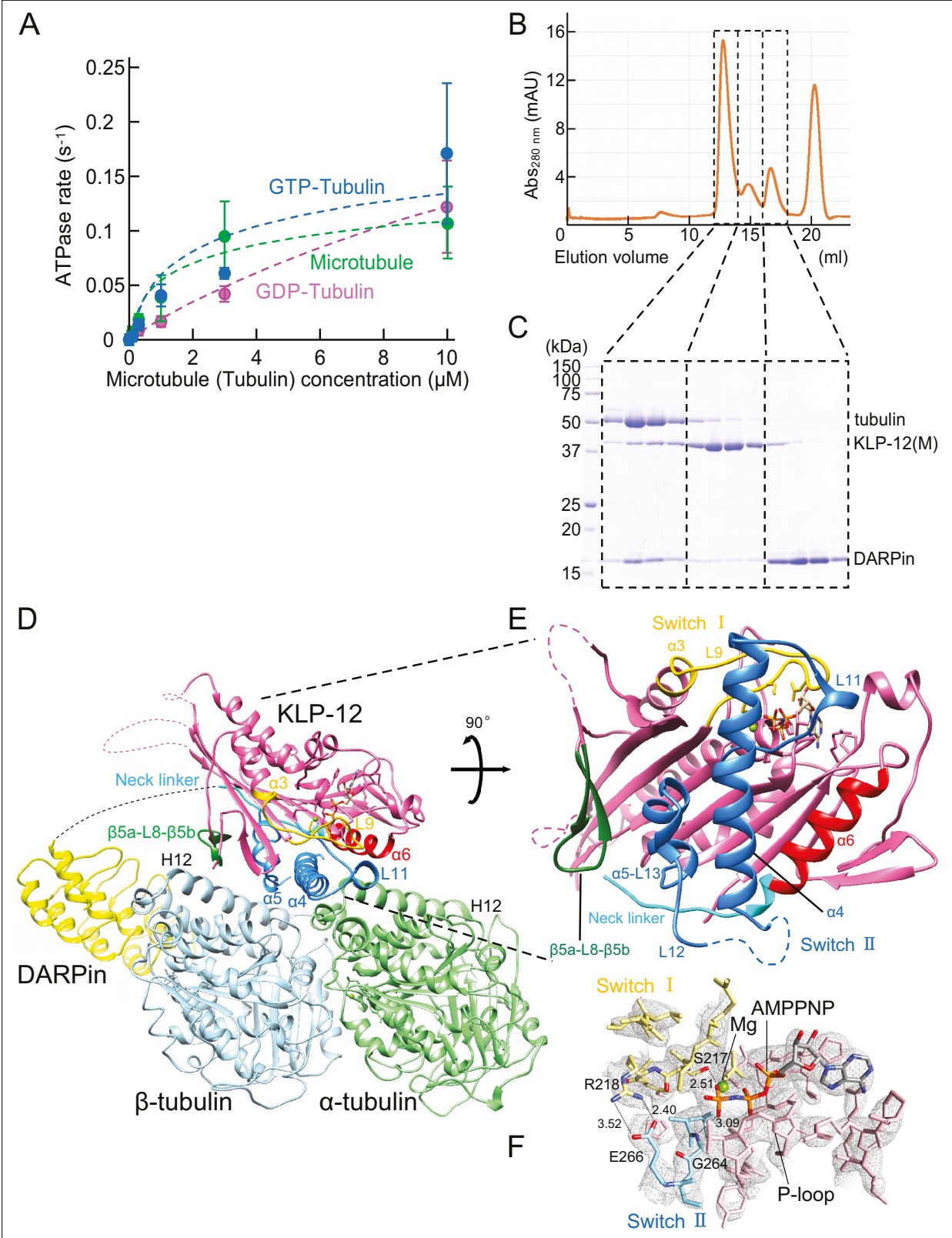

**Figure 3.** Crystal structure of KLP-12-tubulin complex. (**A**) The steady-state ATPase activity of KLP-12(M) measured with GDP tubulin heterodimers, GTP tubulin heterodimers, and microtubules at 30 °C. Error bars represent standard deviation. Tubulin or microtubule GTPase effect was canceled by subtracting control without KLP-12(M). (**B**) Size exclusion chromatography (SEC) of tubulin mixed with KLP-12(M) and DARPin. (**C**) SDS–PAGE analysis of the SEC peaks of tubulin mixed with KLP-12(M) and DARPin. SDS-PAGE analysis of the SEC peaks of tubulin combined with KLP-12–DARPin is

*Figure 3 continued*

available in *Figure 3—figure supplement 2*. (**D**) Crystal structure of the tubulin–KLP-12–DARPin complex. Disordered linkers were drawn as a dotted line. α-tubulin is colored in light green, β-tubulin is in light blue, DARPin is in yellow, and KLP-12 is in magenta. See also *Video 1* for details. (**E**) KLP-12 structure viewed from the interface with tubulin. The residues of the important structure are shown in color. β5a-L8-β5b is green, switch I is yellow, switch II is blue, α6 is red, and the neck linker is cyan. (**F**) Nucleotide binding pocket of KLP-12. The 2Fo-Fc map around AMP-PNP was calculated with coefficient 2Fo− Fc and contoured at 2.0 σ.

The online version of this article includes the following source data and figure supplement(s) for figure 3:

**Source data 1.** The ATPase activity of KLP-12(M) with microtubule.

**Source data 2.** The ATPase activity of KLP-12(M) with GTP tubulin.

**Source data 3.** The ATPase activity of KLP-12(M) with GDP tubulin.

**Source data 4.** The ATPase activity of KLP-12(M).

**Source data 5.** SDS–PAGE gel of the SEC peaks of tubulin mixed with KLP-12(M) and DARPin.

**Figure supplement 1.** ATPase activity of KLP-12–LZ–GFP with microtubule or GTP-Tubulin.

**Figure supplement 1—source data 1.** The ATPase activity of KLP-12–LZ–GFP with microtubule.

**Figure supplement 1—source data 2.** The ATPase activity of KLP-12–LZ–GFP with GTP tubulin.

**Figure supplement 2.** The complex formation of KLP-12–DARPin and tubulin dimers.

**Figure supplement 2—source data 1.** SDS-PAGE gel of the SEC peaks of tubulin mixed with KLP-12–DARPin.

construct) were used to form the tubulin–KLP-12–DARPin complex (*Figure 2A*). Finally, we formed a stable 1:1 complex of KLP-12–DARPin and crystallized GTP-tubulin dimers for structure determination (*Figure 3—figure supplement 2*).

We solved the crystal structure of the GTP-tubulin–KLP-12–DARPin complex at 2.9 Å resolution (*Figure 3D and E*; *Table 1*). KLP-12 forms an ATP conformation in which switches I and II take the closed conformation to hydrolyze ATP, and the neck linker docks to the motor core. The linker between KLP-12 and DARPin, which follows the neck linker, was not observed because of its intrinsic flexibility. In the nucleotide-binding pocket of KLP-12, the density corresponding to AMP-PNP was found with the $Mg^{2+}$ ion (*Figure 3F*). The highly conserved Ser217 of switch I and Gly264 of switch II are coordinated to the γ-phosphate of AMP-PNP. The back door between switch I Arg218 and switch II Glu266 was also closed, representing the pre-hydrolysis state during ATP hydrolysis.

## Structural comparisons among three types of kinesin motors, kinesin-4, kinesin-1, and kinesin-13

To investigate the structural differences of KLP-12 among kinesin-4 subfamily and kinesin superfamily proteins (KIFs), the crystal structure of KLP-12(M) was compared to the structures of the previously reported kinesin-4 members *Mus musculus* KIF4 (PDB ID: 3ZFC) (*Chang et al., 2013*) and KIF7 (PDB ID: 6MLR) (*Jiang et al., 2019*), as well as the well-studied plus-end motor *Homo sapiens* kinesin-1 (KIF5B; PDB ID: 1MKJ, 4HNA, 3J8Y) (*Sindelar et al., 2002*; *Gigant et al., 2013*; *Shang et al., 2014*), and the microtubule destabilizer *Homo sapiens* kinesin-13 (KIF2C; PDB ID: 5MIO) (*Wang et al., 2017*; *Figure 4A*). The degree of similarity was estimated by comparing RMSDs of main chain residues of KLP-12 with other motors (*Figure 4* and *Figure 4—figure supplement 1A*). Note that all structures are in the ATP or ATP-like conformation but structures were determined in different conditions;

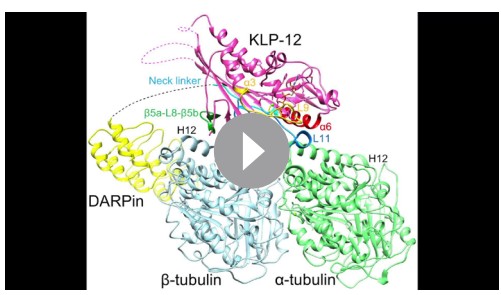

**Video 1.** Crystal structure of KLP-12-tubulin complex.
https://elifesciences.org/articles/77877/figures#video1

KLP-12, KIF5B (4HNA), and KIF2C are bound to tubulin dimers, KIF4 and KIF5B (1MKJ) are the motor domain only, and KIF7 and KIF5B (3J8Y) are bound to the microtubule.

The structure of the KLP-12 (KIF21 subfamily) is very similar to KIF4 among the kinesin-4 family members (*Figure 4B*). Almost no structural differences exist in the main chains, including nucleotide-dependent switch regions, except for the passive conformational changes in α1a and β5a-L8-β5b upon tubulin binding, reflecting the conserved mechanisms of microtubule

**Table 1.** Data collection and refinement statistics.

Tubulin–KLP-12–DARPin

| | | |
|---|---|---|
| Beam line | | SPring-8 BL32XU |
| **Data collection** | | |
| Space group | | $P\,2_1$ |
| Cell dimensions | $a, b, c$ (Å) | 82.33, 80.98, 117.75 |
| | $\alpha, \beta, \gamma$ (°) | 90.00, 93.51, 90.00 |
| Resolution (Å) | | 50–2.88 (3.05–2.88) * |
| $R_{meas}$ (%) | | 35.5 (197.0) |
| $I / \sigma I$ | | 19.28 (3.39) |
| $CC_{1/2}$ | | 97.7 (51.3) |
| Completeness (%) | | 98.2 (98.8) |
| Redundancy | | 87.1 (88.7) |
| **Refinement** | | |
| Resolution (Å) | | 49.25–2.88 |
| No. reflections | | 33796 |
| $R_{work}/R_{free}$ | | 0.211/0.296 |
| No. atoms | Protein | 10301 |
| | Ligand/ion | 93 |
| | Water | 0 |
| B-factors | Protein | 54.67 |
| | Ligand/ion | 42.97 |
| | Water | 0 |
| R.m.s. deviations | Bond lengths (Å) | 0.010 |
| | Bond angles (°) | 1.40 |

*Values in parentheses are for the highest-resolution shell.

dynamics inhibition among kinesin-4 motors (*Bringmann et al., 2004*). On the other hand, KIF7 solved in a complex with microtubule by Cryo-EM shows a moderate structural difference from KLP-12 (*Figure 4A and C*). Switch II (L11-α4-L12-α5-L13) and α6-neck-linker of KIF7 are the most structurally different components from KLP-12 due to the decoupling of switch II conformational change from ATP hydrolysis cycle in KIF7 (*Jiang et al., 2019*). Except for the switch regions, the structures of KLP-12 and KIF7 are very similar, reflecting that they belong to the same kinesin-4 subfamily.

Same as KLP-12, two other motors, KIF5B (4HNA) and KIF2C (5MIO) have been structurally analyzed in a complex with tubulin-dimer in which the exchangeable sites are occupied by the GTP, the GTP analog GDP-AlF₄, and the GDP, respectively (*Figure 4A, B, D and E*). KIF5B binds preferentially to growing GTP microtubules or stabilizes the GDP microtubule (*Muto et al., 2005*; *Peet et al., 2018*; *Shima et al., 2018*), whereas KIF2C destabilizes microtubules (*Ogawa et al., 2004*). KLP-12 takes a similar conformation to KIF5B with RMSD value of 1.898 Å. The structure of the main chain residues facing the tubulin dimer is almost similar, suggesting that the side chain differences should produce distinct functions between KLP-12 and KIF5B (*Figure 4D*). KIF2C, on the other hand, has a very different microtubule-binding interface from KLP-12, including β5a-L8-β5b, switch II, and the class-specific long loop L2 among kinesin-13 (*Figure 4E*). Thus, the conformational differences at the level of main chains should cause the functional differences between KLP-12 and KIF2C.

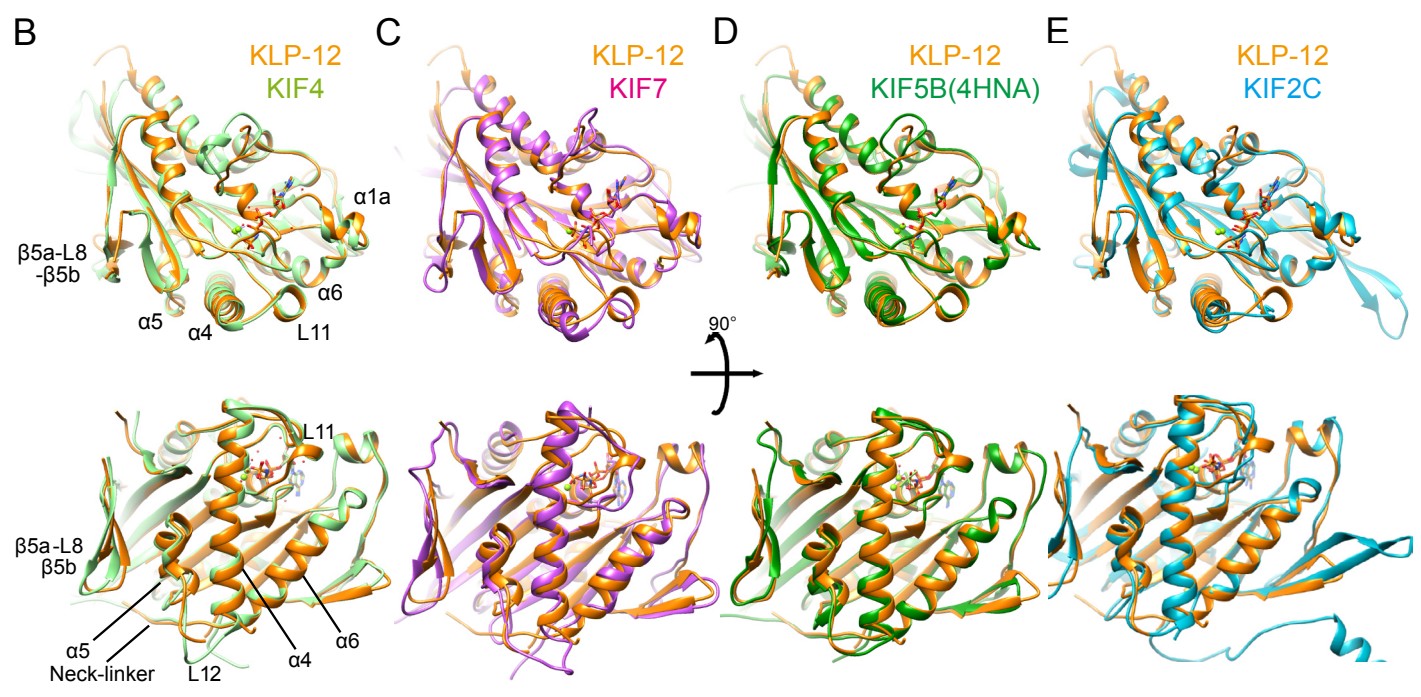

| | Family | PDB ID | Resolution (Å) | RMSD (Å) | No. atoms aligned | Mechanical state | Chemical state |
|---|---|---|---|---|---|---|---|
| KIF4 | kinesin-4 | 3ZFC | 1.80 | 1.928 | 1,292 | Motor only (detached) | AMPPNP |
| KIF7 | kinesin-4 | 6MLR | 4.20 | 3.392 | 1,220 | Microtubule-bound | AMPPNP |
| KIF5B | kinesin-1 | 1MKJ | 2.70 | 2.938 | 1,232 | Motor only (detached) | ADP (ATP-like form) |
| KIF5B | kinesin-1 | 4HNA | 3.19 | 1.927 | 1,280 | GDP-AlF$_4$ tubulin-bound | ADP-AlF$_4$ |
| KIF5B | kinesin-1 | 3J8Y | 5.00 | 1.898 | 1,280 | Microtubule-bound | ATP |
| KIF2C | kinesin-13 | 5MIO | 3.19 | 4.498 | 1,196 | GDP tubulin-bound | AMPPNP |

**Figure 4.** Structural comparison of KLP-12 motor domain with the other kinesins. (**A**) Structural comparison of KLP-12 with the other reported kinesin motor domain structures in various states. (**B**) Superimposition of KLP-12 and KIF4 (Light green). (**C**) Superimposition of KLP-12 and KIF7 (Pink). (**D**) Superimposition of KLP-12 and KIF5B (Green). (**E**) Superimposition of KLP-12 and KIF2C (Cyan).

The online version of this article includes the following figure supplement(s) for figure 4:

**Figure supplement 1.** Structural comparison of KLP-12 motor domain with KIF5B structures.

## Microtubule binding interface of kinesin-4 KLP-12 at α-tubulin

We focused on the interface between α-tubulin and KLP-12 and compared it with the KIF5B and KIF2C structures by superimposing them using the kinesin motor domain (*Figure 5*; *Figure 5—figure supplement 1*). KLP-12 Arg267 interacts with tubulin Glu414/Glu420 in H12 with the hydrophobic support of KLP-12 Val343 in α6 through the hydrophobic stem of α-tubulin Glu420 of α-tubulin, albeit Val343 is not conserved well among kinesin-4 (*Figure 5A and B*). The other interface Lys269 in L11, which is conserved among KIF21 subfamily and KIF4 subfamily but not in KIF7 subfamily, makes a salt-bridge with Glu155 in H4 of α-tubulin. Thus, the KLP-12 and possibly KIF4 generate the triangle contacts among L11 of KLP-12, H4 of α-tubulin, and H12 of α-tubulin. The helix α6 of KLP-12 also supports this interface.

The former Arg267-mediated interaction is conserved in KIF5B through the corresponding residue Lys237 in L11 (*Figure 5B*; *Figure 5—figure supplement 1A*) but not conserved in KIF2C. KIF2C has Ala500 and Glu573 instead of Lys269 and Val343 of KLP-12, resulting in a loss of interaction (*Figure 5B*; *Figure 5—figure supplement 1B*). Instead, KIF2C takes another binding strategy through the KVD finger in loop L2 of KIF2C, as previously reported (*Figure 5—figure supplement 1B*; *Ogawa et al., 2004*; *Trofimova et al., 2018*; *Wang et al., 2017*). The other interface between

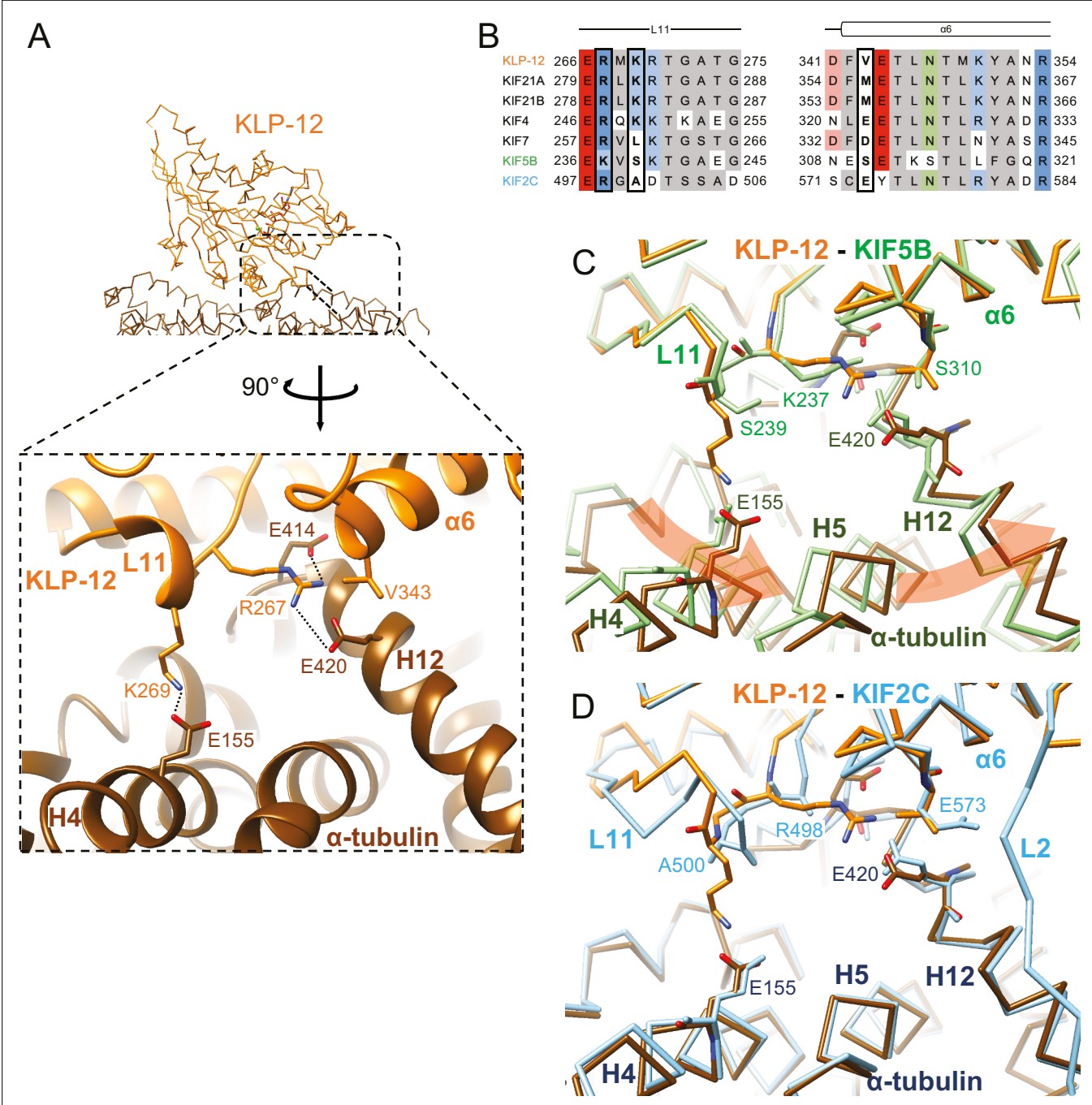

**Figure 5.** Microtubule binding interface of kinesin-4 KLP-12 at α-tubulin. (**A**) KLP-12 Arg267 and Lys269 in loop L11 interact with α-tubulin Glu414 and Glu420 in helix H12 and Glu155 in helix H4, respectively. See *Video 2* for detail. The interfaces of KIF5B and KIF2C at α-tubulin are available in *Figure 4—figure supplement 1*. (**B**) Sequence alignment of the kinesin-4, KIF5B, and KIF2C residues at the interacting area. Interacting residues are marked by squares. (**C**) Superimposition of Cα chain trace models of the KLP-12 complex (orange) and KIF5B complex (green) at kinesin. The rotation direction of α-tubulin between KLP-12 and KIF5B at helices H4, H5, and H12 is illustrated as a red arrow. (**D**) Superimposition of Cα chain trace models of the KLP-12 complex (orange) and KIF2C complex (cyan) at kinesin.

The online version of this article includes the following figure supplement(s) for figure 5:

**Figure supplement 1.** Microtubule binding interface of KIF5B and KIF2C at α-tubulin.

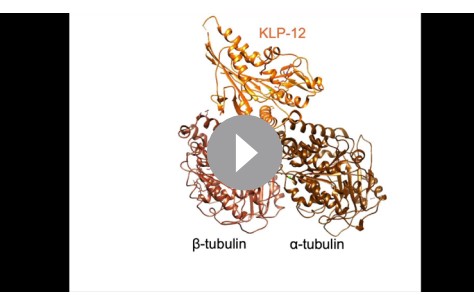

**Video 2.** Microtubule binding interface of kinesin-4 KLP-12 at α-tubulin.
https://elifesciences.org/articles/77877/figures#video2

Lys269 of KLP-12 and Glu155 of α-tubulin is not conserved both in KIF5B and KIF2C. The counterpart residue is Ser239 in KIF5B, which is necessary for the kinesin-1-induced conformational change of microtubules to the growing GTP form (*Shima et al., 2018*; *Figure 5B*; *Figure 5—figure supplement 1A*).

In summary, KLP-12 takes unique triangle contacts with α-tubulin conserved among KIF4 and KIF21 subfamilies. The different interfacial organizations of KLP-12 and KIF5B to α-tubulin result in the counter-clockwise rotation of α-tubulin up to 4 degrees with Glu420 of α-tubulin as the fulcrum, via the formation of a family-specific salt bridge at Lys269-Glu155 (*Figure 5C*). On the other hand, although the interfaces between KLP-12 and KIF2C to α-tubulin are entirely different, the relative binding angle of KLP-12 and KIF2C with α-tubulin is consequently similar (*Figure 5D*). KLP-12, thus, induces a larger rotation of α-tubulin than KIF5B, which builds a configuration of α-tubulin similar to that in the KIF2C-tubulin complex.

## Microtubule binding interface of kinesin-4 KLP-12 at β-tubulin

We next investigated the microtubule-binding interface of KLP-12 at β-tubulin by comparing it to the KIF5B and KIF2C structures superimposed using the kinesin motor domain (*Figure 6*; *Figure 6—figure supplement 1*). Helix α5 of KLP-12 and helix H12 of β-tubulin serve as the interface between KLP-12 and microtubules. Two arginine residues, Arg311 and Arg317, in α5 of KLP-12 form ionic interactions with Glu410 in H12 of β-tubulin (*Figure 6A*). These arginine residues are conserved in KIF5B (*Figure 6B*; *Figure 1—figure supplement 2*); however, they form intramolecular contacts with Glu157 in loop L8 of KIF5B instead of interacting with Glu420 (corresponding to Glu410 in porcine) of β-tubulin (*Figure 6—figure supplement 1A*), suggesting that the organization of α5 of KLP-12 or KIF5B gives rise to the difference.

To further investigate the interactions around α5 of KLP-12 and KIF5B, we carefully observed and found the intramolecular contact between Tyr150 and Asn151 and the resulting additional contact between Asp152 and Lys314 of KLP-12; Asn151 is highly conserved in KIF4 and KIF21 subfamilies and KIF2C, but not in KIF5B (*Figure 6B*; *Figure 1—figure supplement 2*). The corresponding residue of KIF7 is Lys150 similarly acts as asparagine, albeit Leu139 in KIF5B, did not form interaction, therefore Asp140 and Lys281 did not contact (*Figure 6B*; *Figure 6—figure supplement 1A*). The interaction within KLP-12 around Asn151 slightly rearranges the composition of α5 to generate intermolecular interaction between Glu410 of β-tubulin instead of intramolecular interaction with Glu172 of KLP-12. This interacting strategy is basically conserved in KIF2C (*Figure 6B*; *Figure 6—figure supplement 1B*).

In addition to this interaction, KIF2C has another intermolecular contact, which KLP-12 and KIF5B do not have. Arg420 in β5a of KIF2C forms an ionic interaction with Asp414 and Glu417 at the N-terminal side of H12 of β-tubulin (*Figure 6—figure supplement 1B*). This conformation of Arg420 is supported by Leu422 through hydrophobic contact with the side chain stem of Arg420. KLP-12 has His171 instead of Leu422 of KIF2C, showing the repulsive force keeping Arg169, which corresponds to residue Arg420 of KIF2C, away from the acidic residues on H12 (*Figure 6A and B*). KIF5B has Ser154 in the Arg420 position of KIF2C, resulting in no interaction with H12 of β-tubulin (*Figure 6B*; *Figure 6—figure supplement 1A*). These different types of interactions through H12 result in different rotation angles of β-tubulin.

In summary, Arg311 and Arg317 of KLP-12 bridge to Glu410 of β-tubulin produces a clockwise rotation of β-tubulin around the motor domain (*Figure 6C*). KIF5B does not make an equivalent interaction, so the net result is that KLP-12 bends tubulin by 4.6 degrees more than KIF5B. This configuration is possibly conserved in kinesin-4 subfamily, including KIF4 and KIF7, as expected from the amino acid sequence conservations. KIF2C generated further rotation of β-tubulin, resulting in tubulin's highest curvature destabilizing the microtubule end (*Figure 6D*).

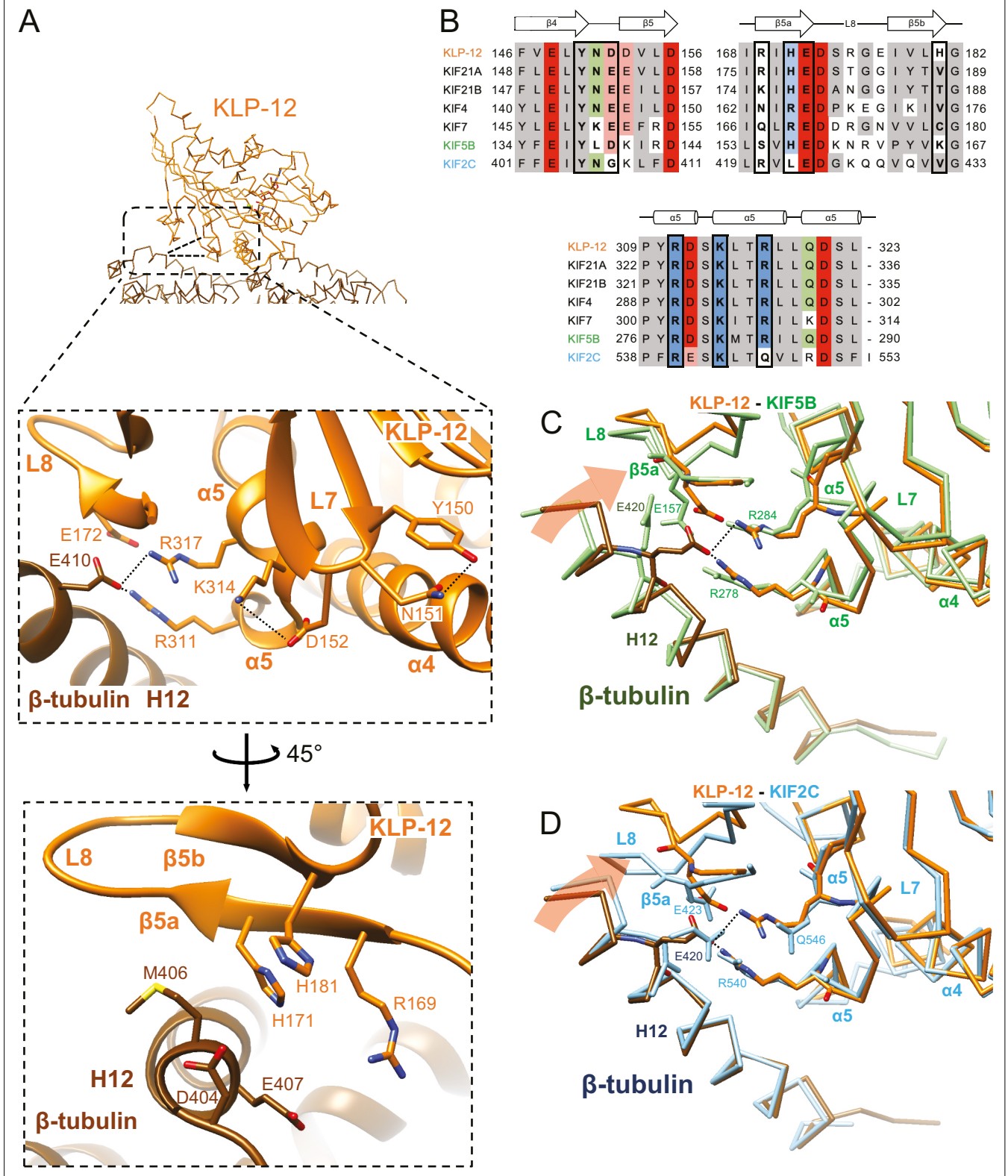

**Figure 6.** Microtubule binding interface of kinesin-4 KLP-12 at β-tubulin helix H12. (**A**) Close-up display of KLP-12 and β-tubulin interaction around β-tubulin helix H12 from the same view as upper panel (middle panel) and 45° rotated view (bottom panel). Glu410 of β-tubulin H12 interacts with Arg311 and Arg317 of KLP-12 helix α5. Tyr150 and Asn151 of KLP-12 form intramolecular interactions. See *Video 3* for detail. The interfaces of KIF5B and KIF2C at β-tubulin are available in *Figure 5—figure supplement 1*. (**B**) Sequence alignment with the secondary structure of the kinesin-4, KIF5B,

*Figure 6 continued on next page*

*Figure 6 continued*

and KIF2C residues at the interacting area. (**C**) Superimposition of Cα chain trace models of the KLP-12 complex (orange) and KIF5B complex (green) at kinesin around β-tubulin H12. (**D**) Superimposition of Cα chain trace models of the KLP-12 complex (orange) and KIF2C complex (cyan) at kinesin around β-tubulin H12.

The online version of this article includes the following figure supplement(s) for figure 6:

**Figure supplement 1.** Microtubule binding interface of KIF5B and KIF2C at β-tubulin helix H12.

## Distinct curvature of tubulin-dimer induced by different kinesin motor domains

As described above, the different types of kinesin motor domains arrange the different kinesin–tubulin interfaces to produce the distinct curvature of tubulin-dimer. To elucidate how kinesin-4 family KLP-12 affects the overall conformation of the tubulin dimer, kinesin–tubulin complex structures were superimposed on α-tubulin and compared (*Figure 7A–C* and *Video 4*). As a result, obvious differences in β-tubulin positions or rotations were observed among the three kinesins. The tubulin dimer with KIF5B, which generates a growing plus-end, forms the straightest conformation. The tubulin dimer with KIF2C, which is a plus-end destabilizing kinesin, forms the most curved conformation, 6.4 degrees larger than that of KIF5B. Intriguingly, KLP-12 induces intermediate curvature of the tubulin dimer, with the α- and β-tubulin angles being 4.6 degrees more curved than KIF5B. This precise control of tubulin curvature by KLP-12 is achieved by counter-clockwise rotation of α-tubulin (*Figure 5C*) and clockwise rotation of β-tubulin (*Figure 6C*) around KLP-12, seen from the left side of the protofilament.

## Discussion

This study investigated the molecular mechanism of microtubule dynamics inhibition by kinesin-4 motor *C. elegans* KLP-12 using genetic, biophysical, biochemical, and structural analyses. *C. elegans* genetics clearly illustrated the role of KLP-12 in regulating the length of axons by inhibiting microtubule dynamics. Biophysical analyses elucidated the plus-end directed motility of KLP-12 and the inhibitory effect of growing microtubules by KLP-12. Biochemical analyses also showed that KLP-12 is similarly active on both the microtubule lattice and the growing microtubule plus-end. Structural studies demonstrated kinesin-4-specific residues at the microtubule-binding interface of KLP-12, adequately increasing the curvature of tubulin-dimers. It might enable the precise curvature of microtubule plus-ends to inhibit microtubule dynamics.

We analyzed two alleles of *klp-12* mutants. Previous studies have shown that the tiling between the PLM axon and ALM cell body is misregulated in *klp-7* mutant worms (*Puri et al., 2021*). *klp-7* is a worm orthologue of KIF2C that has microtubule depolymerizing activity. *klp-7* mutant worms have more stable microtubules and show a larger overlap between PLM axon and ALM cell body. We find similar tiling defects in *klp-12* mutants (*Figure 1*), but the phenotype is weaker than *klp-7*. This could be explained by the activity of KLP-12 in vitro that KLP-12 does not depolymerize microtubules, instead, inhibits the polymerization, unlike KLP-7 (*Figure 2*). Another interesting point is the difference between *klp-12* mutant alleles. A comparison of the null allele *klp-12(tm10890)* and *klp-12(tm5176)*, which has a deletion mutation in the tail-encoding exons, shows milder defects in tiling between ALM and PLM. This would be because *klp-12(tm5176)* is a hypomorphic allele that expresses tail-deleted KLP-12 protein. It is consistent with the previous study showing that full activation of KIF21A, an orthologue of KLP-12, requires binding the tail domain with a cell cortex protein KANK (*van der Vaart et al., 2013*). We need to test the expression of KLP-12 to verify this hypothesis.

We should note here the phenotypic difference between the motor and tail defects; the motor defect made the axon wavy and thinly, whereas the tail defect did not change the thickness of axons (*Figure 1D*). There are two possibilities: (I)

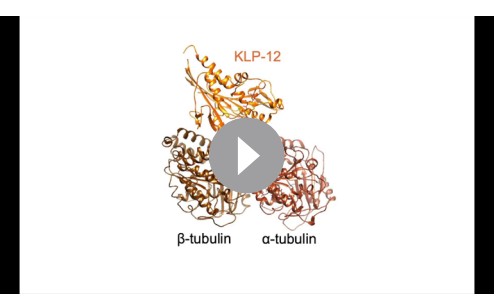

**Video 3.** Microtubule binding interface of kinesin-4 KLP-12 at β-tubulin helix H12.
https://elifesciences.org/articles/77877/figures#video3

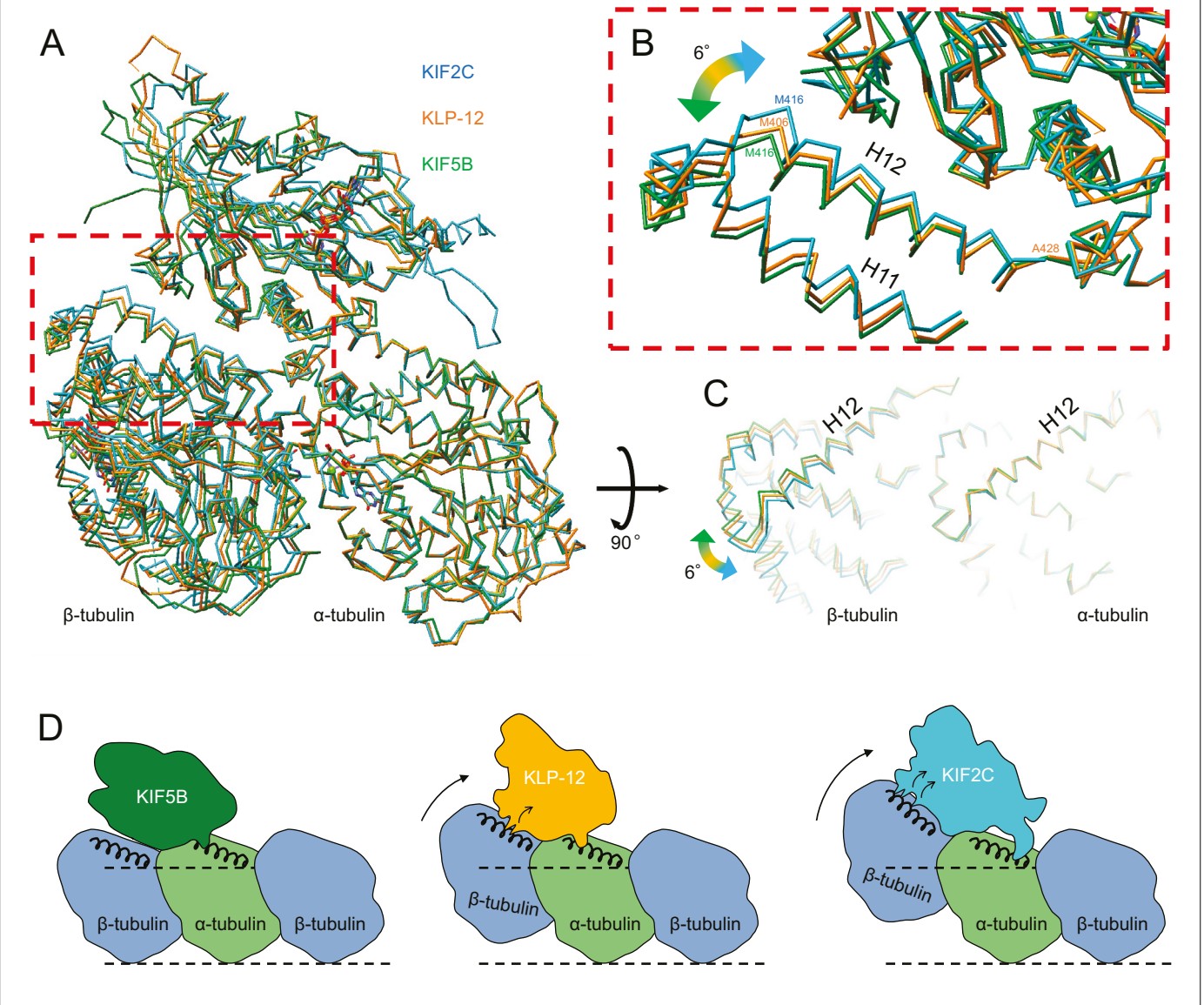

**Figure 7.** Microtubule plus-end control mechanism by kinesin-4. (**A**) Superimposition of KLP-12, KIF5B, and KIF2C complexes at α-tubulin. See *Video 4* for detail. (**B**) Close-up view of superimposed models at the β-tubulin interaction surface. (**C**) Tubulin dimer interface view from kinesin side. (**D**) Models of the microtubule plus-end curving mechanism by kinesins. The kinesin-1 family protein KIF5B does not affect the curvature of microtubules to maintain the conformation of GTPs. Kinesin-4 family protein KLP-12 slightly curves the plus-end by interacting with the mid-portion of β-tubulin H12 to stabilize the plus-end. Kinesin-13 family protein KIF2C strongly curves plus-end by interacting with the N-terminal portion of β-tubulin H12 to destabilize the microtubule.

motor function will be required for proper axon development, or (II) the tail domain without the motor will disable proper axon development. Future studies are required to elucidate this mechanism.

Steady state ATPase assays of the KLP-12 motor domain in the presence of microtubules or GTP-tubulin dimers elucidated the ATPase activities with comparable affinities to microtubules and GTP-tubulin dimers. Although kinesin-13 expresses stronger binding of GDP-tubulin dimers over microtubules, other kinesin motors bind strongly to microtubules over tubulin dimers, including Xklp1 or KIF19A (*Bringmann et al., 2004*; *Hunter et al., 2003*; *Wang et al., 2016*). Therefore, preferential binding to the growing plus-end of microtubules is one of the important features of KLP-12. However, since ATP hydrolysis detaches KLP-12 from the microtubule ends, the tail domain should be required to tether itself to microtubules to continuously stabilize the plus ends.

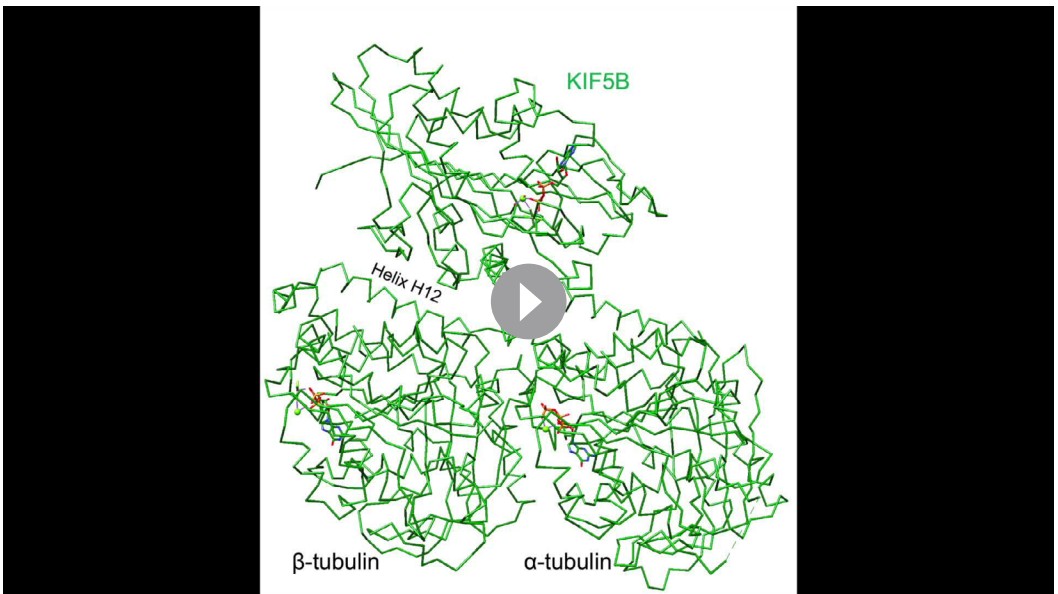

**Video 4.** Microtubule plus-end control mechanism by kinesin-4. Morph movie of KLP-12, KIF5B, and KIF2C. Structures were superimposed by α-tubulin.
https://elifesciences.org/articles/77877/figures#video4

Regarding the ATPase rate, a discrepancy was observed between the single-molecule motility assay and the kinetic study. In the motility assay, the average rate was measured only for motors actively attached to microtubules among the KLP-12 present in the solution. In the kinetics assay, on the other hand, the average speed of all motors on the solution per unit time is calculated. In the former assay, we can detect kinesin-1 movement from 1 nM in solution, whereas KLP-12 movement only from 100 nM or more in the solution. In the latter assay, $K_{M, microtubules}$ of kinesin-1 is only ten times lower than KLP-12 (*Chang et al., 2013*). This difference would be responsible for the discrepancy in the ATPase rate. Considering that SEC and crystallographic analysis show that KLP-12 used in this study is at least structurally uniform (*Figure 3B*), a discrepancy in the ATPase rate may not reflect a difference in microtubule affinity or degradation of motors. Instead, it may come from unknown factors, such as a mixture of motors with an extremely slow or zero ATPase rate.

From the crystal structural analysis, we identified the specific interactions of KLP-12 for α- and β-tubulins (*Figure 4*; *Figure 5*). The contributing residues are mostly conserved among KIF4 and KIF21 subfamilies, suggesting the conserved structural mechanisms. KLP-12 interacts with α-tubulin through the triangle contacts at the N-terminal side of H12 and H4, and β-tubulin at the mid-portion of H12. Kinesin-1 makes single contact with α-tubulin at the N-terminal side of α-tubulin H12 but no contact with β-tubulin. Kinesin-13 binds to α-tubulin at the C-terminal side of H12 through the KVD finger and to β-tubulin at the N-terminal side of H12, in addition to the mid-portion contact. These interactions are summarized in the schematic illustrations in *Figure 7D*.

Kinesin-1 cannot bend the protofilament and allows microtubules to polymerize, whereas kinesin-13 strongly bends the protofilament to depolymerize microtubules (*Ogawa et al., 2004*; *Shima et al., 2018*). KLP-12 expresses the mild effect of protofilament bending, enabling the inhibition of both the polymerization and depolymerization of the microtubules. As summarized previously, changes in the curvature of the peeled plus-ends of microtubules are fundamental to microtubule dynamics (*Brouhard and Rice, 2014*); less curved plus-end polymerizes the microtubule, more curved plus-end shortens the microtubule, and middle curved plus-end stabilizes the microtubule. In addition to the reported structures of KIF5B with tubulin and KIF2C with tubulin representing polymerizing and depolymerizing plus-end of a microtubule, KLP-12 complexed with tubulin structure filled the missing piece of the structural information of stabilizing plus-end of a microtubule. The other domains, including the tail, would further intricately arrange the bending effects of motor domains.

In summary, our study provides important structural clues regarding a novel molecular mechanism by which kinesin-4 inhibits microtubule dynamics. This structural model is consistent with the

previously reported effects of microtubule elongation or destabilization by the motor domains of various kinesin superfamily proteins.

# Materials and methods

## Key resources table

| Reagent type (species) or resource | Designation | Source or reference | Identifiers | Additional information |
|---|---|---|---|---|
| Strain, strain background (*C. elegans*) | Wild type | Obtained from CGC | N2 | |
| Strain, strain background (*C. elegans*) | *uIs31* | Described in *O'Hagan et al., 2005*. | TU2769 | |
| Strain, strain background (*C. elegans*) | *klp-12(tm5176); uIs31* | This study | OTL29 | Deletion mutation in exons encoding the tail domain, generated by Niwa lab. |
| Strain, strain background (*C. elegans*) | *uIs31; jpnEx98[Pmec-7::klp-12]* | This study | OTL135 | Generated by Niwa lab. |
| Strain, strain background (*C. elegans*) | *uIs31; jpnEx98[Pmec-7::klp-12]* | This study | OTL136 | Generated by Niwa lab. |
| Strain, strain background (*C. elegans*) | *klp-12(tm10890); uIs31* | This study | OTL137 | Induces deletion of exon 4–6, resulting in a frameshift, generated by Niwa lab. |
| Recombinant DNA reagent | KLP-12–LZ–GFP | This study | RN349 | pET21B, with C-terminus 6 x His-tag, generated by Nitta lab. |
| Recombinant DNA reagent | KLP-12(M) | This study | RN107 | pGEX-6P, N-terminal GST tag, generated by Nitta lab. |
| Recombinant DNA reagent | DARPin | This study | RN84 | pETDUET-1, with C-terminus 6 x His-tag, generated by Nitta lab. |
| Recombinant DNA reagent | KLP-12–DARPin | This study | RN148 | pGEX-6P, N-terminal GST tag, generated by Nitta lab. |
| Commercial assay or kit | Hi-Fi DNA assembly | NEB | NEB: E2621 | |
| Commercial assay or kit | Glutathione Sepharose 4B | GE Health care | | |
| Software, algorithm | Zoo system | *Hirata et al., 2019* | | |
| Software, algorithm | KAMO | *Yamashita et al., 2018* | | |
| Software, algorithm | XDS | *Kabsch, 2010a* | | |
| Software, algorithm | XSCALE | *Kabsch, 2010b* | | |
| Software, algorithm | PHASER | *McCoy et al., 2007* | | |
| Software, algorithm | SWISS-MODEL | *Waterhouse et al., 2018* | | |
| Software, algorithm | PHENIX | *Liebschner et al., 2017* | | |
| Software, algorithm | UCSF Chimera | *Pettersen et al., 2004* | | |
| Other | Superdex 200 Increase 10/300 GL | GE Health care | | column |

# The amino acid sequence of constructs used in this study

The amino acid sequence of KLP-12–LZ–GFP

MADTCVQVALRIRPQGNREKLEGSRVCTSVLPNDPQ
VTIGGDRSFTYDHVFDMPTLQYVVYESCVEKLVDGL
FDGYNATVLAYGQTGSGKTHTMGTAFDAAVTQKEE
DLGVIPRAIQHTFRKIAECKAQAIEQGLLEPAFEVSV
QFVELYNDDVLDLLSDDRSMSSSIRIHEDSRGEIVL
HGVEQRSVFDMHGTMDILKNGALNRTVAATNMN
EQSSRSHAIFTLHLKQQRVAANPLDESGEQKTGEL
EMEMLCAKFHFVDLAGSERMKRTGATGDRAKEGI
SINVGLLALGNVIAALGGANGKVSHVPYRDSKLTRL
LQDSLGGNSRTLMIACCSPSDSDFVETLNTMKYAN
RAKEIKNKVVANQDKSSKMIGELRSRIAALEAELLE
FKQGKQLEDKVEELASKNYHLENEVARLKKLVMK
DHLIHNHHKHEHAHASKGEELFTGVVPILVELDG
DVNGHKFSVSGEGEGDATYGKLTLKFICTTGKLPV
PWPTLVTTLTYGVQCFSRYPDHMKRHDFFKSAMP
EGYVQERTISFKDDGNYKTRAEVKFEGDTLVNRIE
LKGIDFKEDGNILGHKLEYNYNSHNVYITADKQK
NGIKANFKIRHNIEDGSVQLADHYQQNTPIGDGP
VLLPDNHCLSTQSALSKDPNEKRDHMVLLEFVT
AAGITHGMDELYKLEHHHHHH

The amino acid sequence of KLP-12(M)

MADTCVQVALRIRPQGNREKLEGSRVCTSVLPND
PQVTIGGDRSFTYDHVFDMPTLQYVVYESCVEKL
VDGLFDGYNATVLAYGQTGSGKTHTMGTAFDAA
VTQKEEDLGVIPRAIQHTFRKIAECKAQAIEQGLL
EPAFEVSVQFVELYNDDVLDLLSDDRSMSSSIRIH
EDSRGEIVLHGVEQRSVFDMHGTMDILKNGALN
RTVAATNMNEQSSRSHAIFTLHLKQQRVAANPLD
ESGEQKTGELEMEMLCAKFHFVDLAGSERMKRT
GATGDRAKEGISINVGLLALGNVIAALGGANGKV
SHVPYRDSKLTRLLQDSLGGNSRTLMIACCSPSD
SDFVETLNTMKYANRAKEIKNKVVAN

The amino acid sequence of A-C2 DARPin

MDLGKKLLEAARAGQDDEVRVLMANGADVNA
TDASGLTPLHLAATYGHLEIVEVLLKHGADVSA
SDLMGSTPLHLAALIGHLEIVEVLLKHGADVNA
VDTWGDTPLRLAAVMGHLKIVEALLKHGADVN
AQDKFGKTAYDTSIDNGSEDLAEILQKLNLEHHHHHH

The amino acid sequence of KLP-12–DARPin

MADTCVQVALRIRPQGNREKLEGSRVCTSVLPND
PQVTIGGDRSFTYDHVFDMPTLQYVVYESCVEKL
VDGLFDGYNATVLAYGQTGSGKTHT<u>M</u>GTAFDAA
VTQKEEDLGVIPRAIQHTFRKIAECKAQAIEQGLL
EPAFEVSVQFVELYNDDVLDLLSDDRSMSSSIRIH
EDSRGEIVLHGVEQRSVFDMHGTMDILKNGALN
RTVAATNMNEQSSRSHAIFTLHLKQQRVAANPL
DESGEQKTGELEMEMLCAKFHFVDLAGSERMKR
TGATGDRAKEGISINVGLLALGNVIAALGGANGK
VSHVPYRDSKLTRLLQDSLGGNSRTLMIACCSPS
DSDFVETLNTMKYANRAKEIKNKVVANGGGGSG
GGGSGGGGSGGGGSGGGSDLGKKLLEAARAGQDD
EVRVLMANGADVNATDASGLTPLHLAATYGHLE
IVEVLLKHGADVSASDLMGSTPLHLAALIGHLEIV
EVLLKHGADVNAVDTWGDTPLRLAAVMGHLKIV
EALLKHGADVNAQDKFGKTAYDTSIDNGSEDLAEILQKLNLE

## *C. elegans* experiments

The strains used in this study are described in Key resources table. *C. elegans* strains were maintained as described previously (*Brenner, 1974*). Some strains and OP50 feeder bacteria were obtained from the *C. elegans* genetic center (CGC) (Minneapolis, MN, USA). *klp-12(tm5176)* and *klp-12(tm10890)* were obtained from the National BioResource Project (NBRP, Tokyo, Japan). Transformation of *C. elegans* was performed by DNA injection as described (*Mello et al., 1991*). *uIs31[Pmec-17::GFP]* was used to visualize the morphology of mechanosensory neurons (*O'Hagan et al., 2005*). Worms were fixed using 0.25 mM levamisol (Sigma), 5% agarose pads and 0.1 μm polystyrene beads (Polysciences)

as described (*Niwa, 2017*). Worms were observed by an LSM800 confocal microscope system equipped with a x20 Plan-Apochromat objective lens (NA0.8) (Carl Zeiss).

## Cloning of *klp-12* cDNA

Total worm cDNA was a kind gift from Dr. Kota Mizumoto (University of British Columbia). First, we obtained the full-length *klp-12* cDNA fragment by PCR performed using KOD FX neo (TOYOBO, Tokyo, JAPAN). The first PCR was performed with 5'-gtgaaaATGGCGGACACTTGTGTGC-3' and 5'-ATTACAGAAAAGTGAAAAGGGGTACAAGTG-3' primers, and then the second PCR was performed with 5'-gtgaaaATGGCGGACACTTGTGTGC-3' and 5'-GACAGCATTTGATTTCCAGAATCCGAC–3'. We fully sequenced the fragment and confirmed the sequence of full-length *klp-12* cDNA (NM_001028178). However, we found that the 3' latter half of the *klp-12* cDNA had toxicity in bacteria and could not be inserted into vectors such as pBluescript, pEGFPN1, and pFastbac. Toxicity-resistant strains such as NEB 5-alpha F'I$^q$ (NEB) and ABLE K (Agilent) could not compromise toxicity. Then, based on the sequence data, GeneArt (Thermo Fisher Scientific) was used to synthesize full-length *klp-12* cDNA, which was codon-optimized for *C. elegans*.

## GTP-tubulin preparation

GTP-tubulin was purified from porcine brains following a previously reported method (*Castoldi and Popov, 2003*).

## GDP-tubulin preparation

GDP-tubulin was obtained as the cold disassembly product of microtubules initially polymerized from GTP-tubulin in PEM buffer (0.1 M PIPES-KOH pH 6.8, 1 mM EGTA, 1 mM MgCl$_2$) supplemented with 1 mM GTP at 37 °C for 60 min. After tubulin assembly, microtubules were centrifuged using an Optima TL Ultracentrifuge (Beckman Coulter) with a TLA-100.3 rotor at 48,600×g at 37 °C for 30 min on a cushion containing PEM buffer supplemented with 30% glycerol. The pellet was washed two times with PEM buffer at 37 °C. GDP-tubulin was obtained after dilution in PEM buffer supplemented with 1 mM GDP at 4 °C of pellets.

## Plasmid construction

For preparation of the KLP-12(M) construct, the coding region of KLP-12 (1–365) was further amplified by PCR with an overlapping sequence with pGEX-6P and cloned immediately after the PreScission site of the pGEX-6P vector by Hi-Fi DNA assembly (NEB).

For preparation of the KLP-12–LZ–GFP construct, the N-terminal end of GFP connected to the C-terminal end of KLP-12(1-393) with the leucine zipper (LZ) was amplified by PCR by adding a LZ sequence (*Yue et al., 2018*) and a C-terminal 6 x His tag and cloned into the pET21B vector by Hi-Fi DNA assembly (NEB).

For preparation of the DARPin construct, the coding sequence of A-C2 DARPin was synthesized by the manufacturer (IDT gBlocks). The DNA fragment was amplified by PCR by inserting a C-terminal 6 x His tag and cloned into the pETDUET-1 vector by Hi-Fi DNA assembly (NEB).

For preparation of the KLP-12–DARPin construct, the N-terminal end of DARPin connected to the C-terminal end of KLP-12(M) with the (G$_4$S)$_4$-GGS linker was amplified by PCR by adding a linker sequence and inserted into the pGEX-6P vector by Hi-Fi DNA assembly (NEB).

All constructs used in the study are listed in Key resources table.

## Protein expression and purification

To obtain the KLP-12(M) constructs, the *Escherichia coli* strain BL21(DE3) harboring the plasmid encoding the KLP-12 (1–365) fragment was cultured in LB medium containing 50 µg/ml ampicillin at 37 °C until OD600 >0.4. Protein expression was induced by 0.2 mM IPTG at 18 °C overnight. The cells were harvested by centrifugation and resuspended in 20 mM Tris-HCl pH 8.0 and 171 mM NaCl. The cell suspension was harvested by centrifugation and stored at –80 °C. The cells were resuspended in 50 mM HEPES-KOH pH 7.5, 400 mM KCl, 10% glycerol, 2 mM MgCl$_2$, 5 mM β-mercaptoethanol, and protease inhibitor (0.7 µM leupeptin, 2 µM pepstatin A, 1 mM PMSF, and 2 mM benzamidine). Lysed cells were disrupted using sonication, and the cell debris was removed by centrifugation using an Avanti JXN-30 centrifuge (Beckman Coulter) with a JA-30.50Ti rotor at 80,000×g. The supernatant was

loaded onto 2 ml Glutathione Sepharose 4B (GE Health care) for affinity chromatography, followed by 3 C protease cleavage of glutathione S-transferase (GST). After concentration using an Amicon Ultra concentrator (Merck Millipore; 10 kDa MWCO), the target protein was further purified on a Superdex 200 Increase 10/300 GL size exclusion column (GE Health care) in 20 mM HEPES-KOH pH 7.5, 150 mM NaCl, 2.5 mM MgCl$_2$, and 1 mM DTT. Peak fractions were pooled, concentrated with an Amicon Ultra 10 kDa MWCO concentrator until 10 mg/ml, flash frozen by liquid nitrogen, and stored at –80 °C until use.

To obtain KLP-12–LZ–GFP, the same protocol as for KLP-12(M) was used until the cell suspension was harvested and stored at –80 °C. The cells were resuspended in wash buffer (50 mM HEPES-KOH pH 7.5, 400 mM KCl, 10% glycerol, 2 mM MgCl$_2$, 5 mM imidazole, 5 mM β-mercaptoethanol, and protease inhibitor) and disrupted using sonication. The cell debris was removed by centrifugation using an Avanti JXN-30 centrifuge (Beckman Coulter) with a JA-30.50Ti rotor at 80,000×g. The supernatant was loaded on a 2 ml HIS-Select Nickel Affinity Gel (Merck) and washed with wash buffer. Bound protein was eluted with 50 mM HEPES-KOH pH 7.5, 400 mM KCl, 10% glycerol, 2 mM MgCl$_2$, 300 mM imidazole, 5 mM β-mercaptoethanol, and protease inhibitor. After concentration using an Amicon Ultra 10 kDa MWCO concentrator, the target protein was further purified on a Superdex 200 Increase 10/300 GL size exclusion column (GE Health care) in 20 mM HEPES-KOH pH 7.5, 150 mM NaCl, 2.5 mM MgCl$_2$, and 1 mM DTT. Peak fractions were pooled, and the target protein was further purified on an anion-exchange Mono Q column (GE Health care) with a linear gradient of NaCl in A buffer (20 mM Tris-HCl pH 8.0, 50 mM NaCl, 2.5 mM MgCl$_2$, and 1 mM DTT) and B buffer (20 mM Tris-HCl pH 8.0, 1 M NaCl, 2.5 mM MgCl$_2$, and 1 mM DTT). Peak fractions were pooled, concentrated and buffer changed with an Amicon Ultra 10 kDa MWCO concentrator into A buffer, flash frozen by liquid nitrogen and stored at –80 °C until use.

To obtain the DARPin constructs, the *E. coli* strain BL21(DE3) harboring the plasmid encoding the A-C2 DARPin fragment was cultured in LB medium containing 100 µg/ml ampicillin at 37 °C until OD600 >0.4. Protein expression was induced by 0.1 mM IPTG at 30 °C overnight. The cells were harvested by centrifugation and resuspended in 20 mM Tris-HCl pH 8.0 and 171 mM NaCl. The cell suspension was harvested by centrifugation and stored at –80 °C. The cells were resuspended in wash buffer (50 mM Tris-HCl pH 8.0, 300 mM NaCl, 5 mM imidazole, 5 mM β-mercaptoethanol, and protease inhibitor) and disrupted using sonication. The cell debris was removed by centrifugation using an Avanti JXN-30 centrifuge (Beckman Coulter) with a JA-30.50Ti rotor at 80,000×g. The supernatant was loaded on a 2 ml HIS-Select Nickel Affinity Gel (Merck) and washed with wash buffer. Then, a high salt wash (50 mM Tris-HCl pH 8.0, 1 M NaCl, 5 mM imidazole, 5 mM β-mercaptoethanol, and protease inhibitor) was performed followed by a low salt wash (50 mM Tris-HCl pH 8.0, 10 mM KCl, 5 mM imidazole, 5 mM β-mercaptoethanol, and protease inhibitor). Bound protein was eluted with 50 mM Tris-HCl pH 8.0, 100 mM KCl, 250 mM imidazole, 5 mM β-mercaptoethanol, and protease inhibitor. After concentration using an Amicon Ultra 10 kDa MWCO concentrator, the target protein was further purified on a Superdex 75 10/300 GL size exclusion column (GE Health care) in 20 mM HEPES-KOH pH 7.5, 150 mM NaCl, 2.5 mM MgCl$_2$, and 1 mM DTT. Peak fractions were pooled, concentrated with an Amicon Ultra 10 kDa MWCO concentrator, flash frozen by liquid nitrogen and stored at –80 °C until use.

To obtain KLP-12–DARPin, the same protocol as for KLP-12(M) was used for protein expression and purification, except the use of 100 µg/ml ampicillin for LB medium and 0.1 mM IPTG for protein expression induction.

## Observation of MT dynamics

The in vitro MT observation was performed as reported before (*Al-Bassam, 2014*). Microscopy slides and precleaned glass coverslips were used to assemble a flow chamber using double-sided tape. The chamber was treated with 0.5 mg/ml PLL-PEG-biotin (Surface Solutions, Switzerland) in BRB80 buffer (80 mM K-PIPES, pH 6.8, 1 mM MgCl2, and 1 mM EGTA) for 5 min. After washing the chamber with BRB80 buffer, it was incubated with 0.5 mg/ml Streptavidin for 5 min. Short MT seeds were prepared using 1.5 µM tubulin mix containing 50% biotin-tubulin and 50% AZdye647-tubulin with 1 mM GMPCPP at 37 °C for 30 min. The polymerized MTs were pelleted by ultracentrifuge for 5 min. The pellet was resuspended by BRB80 supplemented with 10% glycerol and fragmented by pipetting. Resultant seeds were aliquoted and snapfrozen by liquid N2. The seeds were melted on 37 °C heat

block immediately before the use and diluted by BRB80. Seeds were attached on the coverslips using biotin-avidin links and incubated with assay buffer (80 mM K-PIPES, pH 6.8, 1 mM MgCl2, and 1 mM EGTA, 0.5% Pluronic F127, 0.1 mg/ml BSA, 0.1 mg/ml biotin-BSA, 0.2 mg/ml k-casein).

The in vitro reaction mixture consisting of 10 µM tubulin, in assay buffer supplemented with 1 mM GTP, oxygen scavenging system composed of Trolox/PCD/PCA, 2 mM ATP, and the specified amount of KLP-12–LZ–GFP was added to the flow chamber. During the experiments the samples were maintained at 30 °C. An ECLIPSE Ti2-E microscope equipped with a CFI Apochromat TIRF 100XC Oil objective lens, an Andor iXion life 897 camera and a Ti2-LAPP illumination system (Nikon, Tokyo, Japan) was used to observe single molecule motility. NIS-Elements AR software ver. 5.2 (Nikon) was used to control the system.

## Single molecule observation

TIRF assay was performed as described before (*Chiba et al., 2019*). To polymerize Taxol-stabilized microtubules labeled with biotin and AZDye647, 30 µM unlabeled tubulin, 1.5 µM biotin-labeled tubulin and 1.5 µM AZDye647-labeled tubulin were mixed in BRB80 buffer supplemented with 1 mM GTP and incubated for 15 min at 37 °C. Then, an equal amount of BRB80 supplemented with 40 µM taxol was added and further incubated for more than 15 min. The solution was loaded on BRB80 supplemented with 300 mM sucrose and 20 µM taxol and ultracentrifuged at 100,000 g for 5 min at 30 °C. The pellet was resuspended in BRB80 supplemented with 20 µM taxol. Polymerized microtubules were flowed into streptavidin adsorbed flow chambers and allowed to adhere for 5–10 min. Unbound microtubules were washed away using assay buffer supplemented with taxol. Purified motor protein was diluted to indicated concentrations in the assay buffer suppelemented with 2 mM ATP and oxygen scavenging system composed of Trolox/PCD/PCA. Then, the solution was flowed into the glass chamber. The sample was observed by the TIRF system as described above.

## ATPase activity

ATPase activity was measured using the EnzChek Phosphate Assay Kit (Molecular Probes) and DS-11 +spectrophotometer (DeNovix). GTP-tubulin was polymerized in PEM buffer (0.1 M PIPES-KOH pH 6.8, 1 mM EGTA, 1 mM MgCl$_2$) supplemented with 1 mM GTP and 4% DMSO (PEM-GTP buffer) at 37 °C for 30 min. When measuring tubulin, the polymerization step was omitted. Each concentration of microtubule or GTP-tubulin was diluted with PEM-GTP buffer. GDP-tubulin was diluted with PEM buffer supplemented with 1 mM GDP (PEM-GDP buffer) to each concentration. The reactions were performed in 5 mM HEPES-KOH pH 7.5, 5 mM potassium acetate, 200 µM MESG, 1.5 U/ml PNP, 10 µM Taxol, and all assays were performed at 30 °C. When measuring as tubulin, Taxol was not added.

### Basal ATPase activity

PEM-GTP buffer was used instead of microtubules (tubulin), no Taxol was added. After 1 mM ATP was added and the absorbance became stable, 1 µM KLP-12(M) was added, and the absorbance was measured every 5 s. It was calculated by subtracting the absorbance of the control measured without KLP-12(M). The experiment was performed three times.

### ATPase activity

After 1 mM ATP was added and the absorbance became stable, 1 µM KLP-12(M) or KLP-12–LZ–GFP was added, and the absorbance was measured every 5 s. It was calculated by subtracting the absorbance of the control measured without KLP-12(M) or KLP-12–LZ–GFP. The experiment was performed three times. Values for $V_{max}$ and $K_m$ were obtained by Lineweaver-Burk plots of ATPase activity versus microtubule or tubulin concentration (1000 nM and 10,000 nM) using Microsoft Excel.

## Size exclusion chromatography analyses

KLP-12–DARPin with tubulin or KLP-12(M), DARPin, and tubulin were analyzed on a Superdex 200 Increase 10/300 GL column equilibrated with 20 mM PIPES-KOH pH 6.8, 50 mM KCl, 1 mM MgCl$_2$, 0.5 mM EGTA, and 10 µM AMP-PNP. Following gel filtration, the proteins were precipitated with acetone, separated by SDS–PAGE, and visualized by Coomassie staining.

## Crystallization and structure determination

Then, 0.6 mg/ml tubulin–KLP-12–DARPin complex, which was purified by size exclusion chromatography, was crystallized at 20 °C by vapor diffusion in crystallization buffer containing 0.2 M ammonium acetate, 0.1 M HEPES pH 7.2–7.8, 22–26% polyethylene glycol (PEG) 3350. Crystals were harvested by Litholoop and transferred into cryoprotectant solution containing 0.2 M ammonium acetate, 0.1 M HEPES pH 7.4, 25% PEG 3350, and 20% glycerol and then flash-frozen in liquid nitrogen. All diffraction datasets were collected at the BL32XU beamline in a synchrotron facility SPring-8 equipped EIGER 9 M detector (DECTRIS Ltd) at –180 °C with wavelengths of 1.00000 Å followed by a ZOO system (*Hirata et al., 2019*). Loop-harvested microcrystals were identified using raster scan and analysis by SHIKA (*Hirata et al., 2019*). Small wedge data, each consisting of 10°, were collected from single crystals, and the collected datasets were processed automatically using KAMO (*Yamashita et al., 2018*) with XDS (*Kabsch, 2010a*), followed by hierarchical clustering analysis using the correlation coefficients of the normalized structure amplitudes between datasets. Finally, a group of outlier-rejected datasets were scaled and merged using XSCALE (*Kabsch, 2010b*). The structure was determined by molecular replacement with the program PHASER (*McCoy et al., 2007*) using the crystal structure of tubulin dimer (PDB ID: 5MIO) as ensemble 1 and the homology model of KLP-12 generated by SWISS-MODEL (*Waterhouse et al., 2018*) with DARPin (PDB ID: 5MIO) as ensemble 2. The electron density map and the structural model were iteratively refined and rebuilt using PHENIX and COOT (*Liebschner et al., 2017*). The Ramachandran statistics are 92.5% and 7.5% in the favored and allowed regions of the Ramachandran plots, respectively, and 0% are outliers. Data collection and refinement statistics are summarized in *Table 1*. All molecular graphics were prepared by using UCSF Chimera (*Pettersen et al., 2004*).

## Acknowledgements

We thank K Chin, T Setsu, and Y Sakihama for assistance and other colleagues for discussions. This work was supported by the Platform Project for Supporting Drug Discovery and Life Science Research (Basis for Supporting Innovative Drug Discovery and Life Science Research (BINDS)) from AMED under Grant Number JP21am0101070. We acknowledge support from the Japan Society for the Promotion of Science (KAKENHI; 19K07246 to T I, 20H03247 and 22H05523 to S N and 19H03396, 21H05254, 21K19352, 22H02795 to R N), AMED-CREST from the Japan Agency for Medical Research and Development (JP21gm0810013 to R N and JP21gm1610003 to T I), JST (Moonshot R&D)(Grant Number JPMJMS2024), FOREST Program (JPMJFR214K to T I), the Takeda Science Foundation to T I and R N, the Mochida Memorial Foundation for Medical and Pharmaceutical Research to T I and R N, the Uehara Memorial Foundation to R N, Bristol-Myers Squibb to R N, and the Hyogo Science and Technology Association to R N.

## Additional information

### Funding

| Funder | Grant reference number | Author |
| --- | --- | --- |
| Japan Society for the Promotion of Science | 19K07246 | Tsuyoshi Imasaki |
| Japan Society for the Promotion of Science | 19H03396 | Ryo Nitta |
| JST (Moonshot R&D) | JPMJMS2024 | Ryo Nitta |
| FOREST Program | JPMJFR214K | Tsuyoshi Imasaki |
| Takeda Science Foundation | | Tsuyoshi Imasaki Ryo Nitta |
| Mochida Memorial Foundation for Medical and Pharmaceutical Research | | Tsuyoshi Imasaki Ryo Nitta |

| Funder | Grant reference number | Author |
| --- | --- | --- |
| Hyogo Science and Technology Association | | Ryo Nitta |
| Japan Society for the Promotion of Science | 22H05523 | Shinsuke Niwa |
| Japan Agency for Medical Research and Development | JP21gm1610003 | Tsuyoshi Imasaki Ryo Nitta |
| Uehara Memorial Foundation | | Ryo Nitta |
| Bristol-Myers Squibb | | Ryo Nitta |
| Japan Science and Technology Agency | | Ryo Nitta |
| Japan Society for the Promotion of Science | 21H05254 | Ryo Nitta |
| Japan Society for the Promotion of Science | 21K19352 | Ryo Nitta |
| Japan Society for the Promotion of Science | 22H02795 | Ryo Nitta |
| Japan Society for the Promotion of Science | 20H03247 | Shinsuke Niwa |
| Japan Agency for Medical Research and Development | JP21gm0810013 | Tsuyoshi Imasaki Ryo Nitta |

The funders had no role in study design, data collection and interpretation, or the decision to submit the work for publication.

## Author contributions

Shinya Taguchi, Conceptualization, Data curation, Formal analysis, Investigation, Writing – original draft, Writing – review and editing; Juri Nakano, Tomoki Kita, Formal analysis, Investigation; Tsuyoshi Imasaki, Shinsuke Niwa, Conceptualization, Supervision, Investigation, Writing – original draft, Writing – review and editing; Yumiko Saijo-Hamano, Naoki Sakai, Hideki Shigematsu, Hiromichi Okuma, Takahiro Shimizu, Eriko Nitta, Satoshi Kikkawa, Satoshi Mizobuchi, Investigation; Ryo Nitta, Conceptualization, Supervision, Writing – original draft, Project administration, Writing – review and editing

## Author ORCIDs

Shinya Taguchi http://orcid.org/0000-0002-9868-0651
Tsuyoshi Imasaki http://orcid.org/0000-0001-5462-1820
Hideki Shigematsu http://orcid.org/0000-0003-3951-8651
Shinsuke Niwa http://orcid.org/0000-0002-8367-9228
Ryo Nitta http://orcid.org/0000-0002-6537-9272

## Decision letter and Author response

Decision letter https://doi.org/10.7554/eLife.77877.sa1
Author response https://doi.org/10.7554/eLife.77877.sa2

# Additional files

## Supplementary files
• Transparent reporting form

## Data availability
Diffraction data have been deposited in PDB under the accession code 7X4N.

The following dataset was generated:

| Author(s) | Year | Dataset title | Dataset URL | Database and Identifier |
|---|---|---|---|---|
| Taguchi S, Imasaki T, Nitta R | 2022 | Structural model of microtubule dynamics inhibition by kinesin-4 from the crystal structure of KLP-12 -tubulin complex | https://www.rcsb.org/structure/7X4N | RCSB Protein Data Bank, 7X4N |

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
