## [Editor Report]

In their study, Taguchi et al. aim to determine how a member of the kinesin-4 family is able to stabilize the tips of microtubules to suppress both their growth and shrinkage, a process important for normal development. This paper provides convincing data on KLP-12 by combining in vivo *C. elegans* work with in vitro single-molecule analysis and structural studies of the motor domain. The structure shows that KLP-12 bends tubulin heterodimers to a level that lies in between the extremes of bending by KIF5B (lattice stabilizer) and KIF2C (lattice destabilizer). This important study will be of interest to those in the fields of neuronal development and cytoskeletal dynamics.

---

## [Decision Letter]

**Decision letter after peer review:**

Thank you for submitting your article "Structural model of microtubule dynamics inhibition by Kinesin-4 from the crystal structure of KLP-12 -tubulin complex" for consideration by *eLife*. Your article has been reviewed by 3 peer reviewers, and the evaluation has been overseen by a Reviewing Editor and Suzanne Pfeffer as the Senior Editor. The following individuals involved in the review of your submission have agreed to reveal their identity: Robert A Cross (Reviewer #1); Jing Xu (Reviewer #2); Khanh Huy Bui (Reviewer #3).

Essential revisions:

All three reviewers agreed that this is a robust and thorough study that warrants publication in *eLife* once it has been revised with additional analyses and textual changes. While additional analyses from the in vitro assays constitute the major revision necessary for publication, please go through the reviewer comments carefully and revise and expand the text as necessary. There are multiple areas where additional details or discussion can benefit the paper and other areas where claims should be tempered based on the data presented.

Specifically, all three reviewers noted that they would like to see additional analyses of KLP-12 in the in vitro TIRF studies. First, does KLP-12 accumulate at the plus ends of the microtubules, either in the TIRF assay with taxol-stabilized microtubules, or in the dynamics assay? It is also important to show the localization of KLP-12 in Figure 2E. If the concentration is too high, the authors should drop the concentration as suggested by Reviewer 3. Second, there are many other parameters that the authors can quantify in their dynamic microtubule assay. Please refer to the comments from all three reviewers to bolster this section. Based on the n reported in the figure legend, the authors should be able to review their data and provide additional analyses and details that would strengthen the paper greatly without performing additional experiments.

*Reviewer #1 (Recommendations for the authors):*

This is a comprehensive effort to find out how members of the kinesin-4 family are able to stabilise microtubule tips so as to suppress both growth and shrinkage. Using crystallography of soluble tubulin in complex with KLP-12, a kinesin-4 from *C. elegans*, electron microscopy, and biochemistry, together with cell biological characterisation of mutants in cultured neurons, the authors argue that kinesin-4 motors exert their effects by stabilising tubulin in a bent conformation that is too straight to depolymerise readily and too bent to insert stably into the lattice.

The work is interesting but I do have some comments, mostly about the clarity of presentation and argument.

1. It is proposed that all kinesins-4 make the same interface with tubulin, yet some are motile and others are not. This profound difference is mentioned but little discussed, the focus is all on the ability of kinesins-4 to stably curve tubulin. Yet the ability to bind reversibly to the unbent GDP lattice is also central to (motile) function.

2. Processive KIF12 LZ dimers move at a kinesin-1-like rate of 800 nm per sec. Assuming one ATP is consumed per step, this implies an ATPase of ~50 per sec per head (50 * 2 * 8nm per sec). The reported maximum monomer rate of 0.16 s^-1^ is wildly inconsistent with the dimer walking rate. This is not discussed. The observed 1000x activation is consistent with previous work on kinesin-1.

3. Does GFP-KLP12 LZ enrich at plus ends? I was hoping to see data from the GFP KIF12 LZ construct in TIRF assays. The walking action of KLP12 could be very important to deliver it to microtubule tips because the KLP12 motor domains prefer free tubulin, which will ordinarily be present at a much greater concentration than that in the splayed protofilaments at the microtubule tips.

4. GTP tubulin was used in the ATPase assays, but not GDP tubulin. It would be helpful to know if GDP tubulin activates KIF12.

5. The core of the paper is the new crystal structure with GMPCPP tubulin whose assembly is blocked by DARPIN. As the authors discuss, the new tubulin-KIF12 structure is very similar to the 4HNA (KIF5B.ADP.ALF4) structure. The rms deviation is similar to that between KIF4 and KIF12. The interface with tubulin is very similar. Yet KIF5B is activated by microtubules and not free tubulin, whereas KIF-12 prefers free tubulin. This seems like a problem for the authors' argument – since the data say that despite the kinesin-4 family-specific features of the interface, the KIF-12-tubulin structure is overall very similar indeed to the KIF5B-tubulin structure?

6. The comparison of the KLP12 interface to KIF5B and KIF2C is helpful, but hard to follow. The ms lacks a clear summary statement of what are the conserved, kinesin-4 family-specific features of the interface. The claim is that KLP12 bends tubulin slightly more than KIF5B, by rotating α-tubulin by 4 degrees, with E420 of α-tubulin as the fulcrum, via the formation of a family-specific salt bridge at K269-E155. In β-tubulin, R311 and R317 bridge to E410, also producing a rotation. KIF5B does not make an equivalent interaction. The net result is that KLP12 bends tubulin by 3 degrees more than KIF5B. Please clarify that this is compatible with the earlier-discussed high similarity of the current structure to 4HNA. Please clarify that the 3 kinesins all rotate tubulin in the same plane. Relatedly it might be helpful to provide a morph animation of tubulin in the three complexes.

7. Of less concern, there are some typos and language glitches.

L461 'builds a composition of KIF2C similar to that of α tubulin'.

I think what is meant is 'a configuration of α-tubulin similar to that in the KIF2C complex'.

L534 'Glowing plus end' should be 'growing plus end'.

L585 'turn the pealed end into a workspace' – don't understand. Possibly should be 'target KLP12 to protofilament peels'.

*Reviewer #2 (Recommendations for the authors):*

– Figure 1C: Please explain what AVM and PVM are. ALM and PLM are explained in the caption for panel D.

– Figure 1D & E: Do larger overlaps in ALM indicate longer neurons? What is the sample size?

– Figure 1G: Please indicate the sample size, what the horizontal bars in the graph are, what statistical analysis was used, and what the associated P-value is.

– Figure 2C & D: Please clarify the data analysis methods used here. How are the mean and errors determined here?

– Figure 2D: The authors may consider reducing the range of run-length shown in the graph.

– Figure 2 —figure supplement 1: Please clarify what the dotted lines are.

– Line 385: "… reflecting the conserved mechanisms of the microtubule-stabilizing effect of kinesin-4".

Inhibiting microtubule dynamics is not necessarily the same as stabilizing microtubules. Do the authors have data demonstrating that kinesin-4 stabilizes microtubules?

-- Lines 613-614: "slightly more curving inhibits plus-end directed motility along the microtubule lattice".

Do the authors have biophysics data demonstrating that plus-end directed motility is inhibited by slightly more curved tubulin?

*Reviewer #3 (Recommendations for the authors):*

1) In Figure 1, two mutants were studied leading to a phenotype. Could the authors explain in more detail what mutations were made and why, as well as their predicted effects on the protein structure/function? Also, is there a way to verify that the mutant proteins are indeed expressed in truncated form or completely missing?

2) Microtubule dynamics are generally assessed using four parameters: growth rate, depolymerization rate, catastrophe frequency and rescue frequency. What is the effect of KLP-12-LZ-GFP on the parameters that were not measured? Also, since KLP-12 decreases growth rate, does it also decrease the length of microtubules?

3) What is the behaviour of KLP-12-LZ-GFP on dynamic microtubules? In figure 2B, which was done on taxol-stabilized microtubules, the motors seem to fall off the microtubule at the end. Alternatively, could the authors discuss how they think the motor decreases growth rate by walking along a microtubule and falling off it at the end?

4) Figure 2 E only shows 1 concentration, can the author show more than 1 concentration?

5) I wonder whether there is any mutation which is mentioned in the literature that can be mapped onto the motor domain?

---

## [Author Response]

Essential revisions:All three reviewers agreed that this is a robust and thorough study that warrants publication in eLife once it has been revised with additional analyses and textual changes. While additional analyses from the in vitro assays constitute the major revision necessary for publication, please go through the reviewer comments carefully and revise and expand the text as necessary. There are multiple areas where additional details or discussion can benefit the paper and other areas where claims should be tempered based on the data presented.Specifically, all three reviewers noted that they would like to see additional analyses of KLP-12 in the in vitro TIRF studies. First, does KLP-12 accumulate at the plus ends of the microtubules, either in the TIRF assay with taxol-stabilized microtubules, or in the dynamics assay?

We thank the reviews for suggesting a critical experiment. We performed the TIRF assay and observed the behavior of KLP-12 on the taxol-stabilized and dynamic microtubules. In our observation, we have not seen KLP-12 accumulating at the tip of microtubules. This is consistent with the results of KIF21A/B, a mammalian orthologue of KLP-12 (Vaart et al., Dev Cell, 2013; van Riel et al., *eLife*, 2017). We added these data into the revised Figure 2.

It is also important to show the localization of KLP-12 in Figure 2E. If the concentration is too high, the authors should drop the concentration as suggested by Reviewer 3.

We revised the figure to add the localization of KLP-12 (revised Figure 2E).

Second, there are many other parameters that the authors can quantify in their dynamic microtubule assay. Please refer to the comments from all three reviewers to bolster this section. Based on the n reported in the figure legend, the authors should be able to review their data and provide additional analyses and details that would strengthen the paper greatly without performing additional experiments.

We thank reviewers for suggesting important points. We performed additional experiments and added new data showing the microtubule dynamics (Figure 2E-I). The microtubule growth rate was reduced in the presence of KLP-12, whereas the microtubule depolymerization rate was not affected (Figures 2F and 2H). The frequency of catastrophe events was slightly reduced with KLP-12 (Figure 2I). This is similar to the previous results of KIF21A- or KIF5-bound microtubules, thus, this property is conserved in a broad range of kinesins. The frequency of rescue events was reduced as well (Figure 2G). It rises two possibilities; one is that KLP-12 suppresses microtubule polymerization, and another is the indirect effect induced by the reduction of MT catastrophe events. We have included these in the result section (page 9, line 202-204; Figure 2).

Reviewer #1 (Recommendations for the authors):This is a comprehensive effort to find out how members of the kinesin-4 family are able to stabilise microtubule tips so as to suppress both growth and shrinkage. Using crystallography of soluble tubulin in complex with KLP-12, a kinesin-4 from *C. elegans*, electron microscopy, and biochemistry, together with cell biological characterisation of mutants in cultured neurons, the authors argue that kinesin-4 motors exert their effects by stabilising tubulin in a bent conformation that is too straight to depolymerise readily and too bent to insert stably into the lattice.The work is interesting but I do have some comments, mostly about the clarity of presentation and argument.1. It is proposed that all kinesins-4 make the same interface with tubulin, yet some are motile and others are not. This profound difference is mentioned but little discussed, the focus is all on the ability of kinesins-4 to stably curve tubulin. Yet the ability to bind reversibly to the unbent GDP lattice is also central to (motile) function.

We thank this reviewer for pointing out the critical issue. We have compared our KLP-12 structure with other reported kinesin-4 structures (revised Figure 4). In summary, KLP-12 (KIF21 subfamily) and KIF4 (KIF4 subfamily) take very similar structures, and critical residues for microtubule-end binding are highly conserved between them. These residues are partially conserved in KIF7, albeit switch II conformation of KIF7 is different from KIF4 and KIF21 due to the decoupling of switch II from switch I (nucleotide-binding pocket). It results in loss of motility in KIF7 and the behavior of the stay in the plus-end of the microtubule to inhibit dynamics (Jiang et al., Dev. Cell 2019). We have included these in the result section (pages 13-14, lines 321-344; Figure 2).

2. Processive KIF12 LZ dimers move at a kinesin-1-like rate of 800 nm per sec. Assuming one ATP is consumed per step, this implies an ATPase of ~50 per sec per head (50 * 2 * 8nm per sec). The reported maximum monomer rate of 0.16 s^-1^ is wildly inconsistent with the dimer walking rate. This is not discussed. The observed 1000x activation is consistent with previous work on kinesin-1.

We think the reason is because of the less binding of KLP-12 to the microtubule (See Author response image 1). At 50 nM, KIF1A(1-393)::mSca (KIF1A motor domain fused with mScarlet) fully decorates microtubules. In contrast, the binding of KLP-12-LZ-GFP (KLP-12(1-393)LZ::GFP in the figure) sparsely binds to microtubules. This property is similar to KIF21A/B (Vaart et al., Dev Cell, 2013; van Riel et al., *eLife*, 2017). We described this discrepancy in the result session (page 11 lines 259-260) and added the discussion to the Discussion section (pages 24-25, lines 580-592).

**Author response image 1. sa2fig1:** 

3. Does GFP-KLP12 LZ enrich at plus ends? I was hoping to see data from the GFP KIF12 LZ construct in TIRF assays. The walking action of KLP12 could be very important to deliver it to microtubule tips because the KLP12 motor domains prefer free tubulin, which will ordinarily be present at a much greater concentration than that in the splayed protofilaments at the microtubule tips.

As shown in revised Figure 2E, KLP-12 did not accumulate to microtubule tips. This is consistent with the data from KIF21A/B (Vaart et al., Dev Cell, 2013; van Riel et al., *eLife*, 2017).

4. GTP tubulin was used in the ATPase assays, but not GDP tubulin. It would be helpful to know if GDP tubulin activates KIF12.

We thank this reviewer for this critical suggestion. We did an ATPase assay with GDP tubulin and found it has almost ten times weaker affinity than GTP-tubulin or microtubule (revised Figure 3A). Hence, KLP-12 selectively binds to the growing GTP microtubule to suppress the microtubule growth from the plus-end. This result is consistent with the new TIRF data in revised Figure 2.

5. The core of the paper is the new crystal structure with GMPCPP tubulin whose assembly is blocked by DARPIN. As the authors discuss, the new tubulin-KIF12 structure is very similar to the 4HNA (KIF5B.ADP.ALF4) structure. The rms deviation is similar to that between KIF4 and KIF12. The interface with tubulin is very similar. Yet KIF5B is activated by microtubules and not free tubulin, whereas KIF-12 prefers free tubulin. This seems like a problem for the authors' argument – since the data say that despite the kinesin-4 family-specific features of the interface, the KIF-12-tubulin structure is overall very similar indeed to the KIF5B-tubulin structure?

As this reviewer pointed out, the rmsd of the “main chain” residues between KLP-12 and KIF5B is similar. However, if we see side chains, they had a clear difference which changes the binding properties to the GTP tubulin-dimers. We clarified these points in the revised manuscript (page 15, lines 371-374).

6. The comparison of the KLP12 interface to KIF5B and KIF2C is helpful, but hard to follow. The ms lacks a clear summary statement of what are the conserved, kinesin-4 family-specific features of the interface. The claim is that KLP12 bends tubulin slightly more than KIF5B, by rotating α-tubulin by 4 degrees, with E420 of α-tubulin as the fulcrum, via the formation of a family-specific salt bridge at K269-E155. In β-tubulin, R311 and R317 bridge to E410, also producing a rotation. KIF5B does not make an equivalent interaction. The net result is that KLP12 bends tubulin by 3 degrees more than KIF5B. Please clarify that this is compatible with the earlier-discussed high similarity of the current structure to 4HNA. Please clarify that the 3 kinesins all rotate tubulin in the same plane. Relatedly it might be helpful to provide a morph animation of tubulin in the three complexes.

We thank this reviewer for the better presentation of our structure. As this reviewer suggested, we have revised the text (page 18, lines 428-432, page 20, 476-481). We also made a morph animation of comparison of KLP-12, KIF5B, and KIF2C (Movies 1-4).

7. Of less concern, there are some typos and language glitches.L461 'builds a composition of KIF2C similar to that of α tubulin'.I think what is meant is 'a configuration of α-tubulin similar to that in the KIF2C complex'.

This reviewer is correct and we have revised the text as this reviewer suggested (page 18, lines 435-436).

L534 'Glowing plus end' should be 'growing plus end'.

We corrected the typo.

L585 'turn the pealed end into a workspace' – don't understand. Possibly should be 'target KLP12 to protofilament peels'.

We rewrote this sentence (page 24, lines 573-576).

Reviewer #2 (Recommendations for the authors):– Figure 1C: Please explain what AVM and PVM are. ALM and PLM are explained in the caption for panel D.

AVM (anterior ventral mechanosensory) and PVM (posterior ventral mechanosensory) neurons are mechanosensory neurons. We added the explanation into the caption for Figure 1C.

– Figure 1D & E: Do larger overlaps in ALM indicate longer neurons? What is the sample size?

Neurons may not necessarily be getting longer; for example, the length of the worm could be changing. What we can safely say here is that the overlap is increasing. Larger overlap is one of the phenotypes caused by more stable microtubules. For example, a larger overlap is observed in kinesin-13 (klp-7) mutant worms. We described it in the discussion (pages 21-22, lines 517-533). We inserted 50 µm scale bars into figures 1D and 1F.

– Figure 1G: Please indicate the sample size, what the horizontal bars in the graph are, what statistical analysis was used, and what the associated P-value is.

We have included sample size, P value, and statistical method in the figure legend. We also included what the bars represent.

– Figure 2C & D: Please clarify the data analysis methods used here. How are the mean and errors determined here?

We have included the data obtained by taxol-stabilized microtubules and dynamic microtubules (Revised Figures 2D and E). Mean ± standard deviation is written in the Figure legends. Statistical methods are also written in the Figure legend.

– Figure 2D: The authors may consider reducing the range of run-length shown in the graph.

We have revised the graph and showed the run-length (revised Figure 2E).

– Figure 2 —figure supplement 1: Please clarify what the dotted lines are.

The dotted lines are an exponential trend line of the plot. We wrote it in the figure legend.

– Line 385: "… reflecting the conserved mechanisms of the microtubule-stabilizing effect of kinesin-4".Inhibiting microtubule dynamics is not necessarily the same as stabilizing microtubules. Do the authors have data demonstrating that kinesin-4 stabilizes microtubules?

This reviewer is correct and we revised the explanation (page 14, lines 329-331).

-- Lines 613-614: "slightly more curving inhibits plus-end directed motility along the microtubule lattice".Do the authors have biophysics data demonstrating that plus-end directed motility is inhibited by slightly more curved tubulin?

We removed this sentence in the revised manuscript due to excessive wording.

Reviewer #3 (Recommendations for the authors):1) In Figure 1, two mutants were studied leading to a phenotype. Could the authors explain in more detail what mutations were made and why, as well as their predicted effects on the protein structure/function?

The mutant is following; tm10890 is a mutant that induces deletion of the motor domain. Because the mutation causes frameshift, tm10890 should be a null allele. A mutant tm5176 induces deletion of WD repeats. As tm5176 deletes the C-terminal region, the mutant should express deletion mutant. We described these explanations in the legend for Figure 1B.

Also, is there a way to verify that the mutant proteins are indeed expressed in truncated form or completely missing?

We need some antibodies to determine whether these mutants are totally null or a truncated form of the mutant protein is expressed. Because the focus of this study is the structure of KLP-12, we would like to reserve this point as a discussion.

2) Microtubule dynamics are generally assessed using four parameters: growth rate, depolymerization rate, catastrophe frequency and rescue frequency. What is the effect of KLP-12-LZ-GFP on the parameters that were not measured?

We thank this reviewer for pointing out the important issue. We found growth rate is the most affected parameter. Depolymerization rate was not significantly affected and the frequency of MT catastrophe events was slightly reduced (Figure 2H). This is similar to the result of KIF21A- or KIF5-bound microtubules. Thus, the property is conserved in a broad range of kinesins. Frequency of rescue events was reduced as well (Figure 2H). One possibility is that KLP-12 suppresses microtubule polymerization. Another possibility is the indirect effect induced by reduced MT catastrophe events. We have included these in the result section and Discussion section.

Also, since KLP-12 decreases growth rate, does it also decrease the length of microtubules?

We think it is difficult to quantitatively compare the length of microtubules in vitro because of the following reasons; (1) the length of GMPCPP seeds varies (2) the length of polymerized microtubules strongly depends on the duration of polymerization and (3) the length of microtubules polymerized from seeds is not constant because of dynamic instability. We think the growth rate is a better parameter that is not affected by these variations (Figure 2F).

3) What is the behaviour of KLP-12-LZ-GFP on dynamic microtubules? In figure 2B, which was done on taxol-stabilized microtubules, the motors seem to fall off the microtubule at the end. Alternatively, could the authors discuss how they think the motor decreases growth rate by walking along a microtubule and falling off it at the end?

We added the motility of KLP-12-LZ-GFP on dynamic microtubules. We found KLP-12-LZ-GFP does not stay at the tip of microtubules and falls off at the end.

4) Figure 2 E only shows 1 concentration, can the author show more than 1 concentration?

We showed two concentrations.

5) I wonder whether there is any mutation which is mentioned in the literature that can be mapped onto the motor domain?

We did not know of any other mutation so far.